# Unveiling aerosol-cloud interactions Part 1: Cloud contamination in satellite products enhances the aerosol indirect forcing estimate

Matthew W. Christensen[1,2], David Neubauer[3], Caroline A. Poulsen[1], Gareth E. Thomas[1], Gregory R. McGarragh[2], Adam C. Povey[4], Simon R. Proud[2], and Roy G. Grainger[4]

[1]RAL Space, STFC Rutherford Appleton Laboratory, Harwell, OX11 0QX, United Kingdom
[2]Atmospheric, Oceanic and Planetary Physics, University of Oxford, Oxford, OX1 3PU, United Kingdom
[3]Institute for Atmospheric and Climate Science, ETH Zurich, Zurich, 8092, Switzerland
[4]National Centre for Earth Observation, University of Oxford, Oxford, OX1 3PU, United Kingdom

*Correspondence to:* Matthew Christensen (Matthew.Christensen@stfc.ac.uk)

**Abstract.** Increased concentrations of aerosol can enhance the albedo of warm low-level cloud. Accurately quantifying this relationship from space is challenging due in part to contamination of aerosol statistics near clouds. Aerosol retrievals near clouds can be influenced by stray cloud particles in areas assumed to be cloud-free, particle swelling by humidification, shadows and enhanced scattering into the aerosol field from (3D radiative transfer) clouds. To screen for this contamination we have developed a new Cloud-Aerosol Pairing Algorithm (CAPA) to link cloud observations to the nearest aerosol retrieval within the satellite image. The distance between each aerosol retrieval and nearest cloud is also computed in CAPA.

Results from two independent satellite imagers, the Advanced Along Track Scanning Radiometer (AATSR) and MODerate Resolution Imaging Spectroradiometer (MODIS) show a marked reduction in the strength of the intrinsic aerosol indirect radiative forcing when selecting aerosol pairs that are located farther away from the clouds ($-0.28 \pm 0.26$ W/m$^2$) compared to those including pairs that are within 15 km of the nearest cloud ($-0.49 \pm 0.18$ W/m$^2$). The larger aerosol optical depths in closer proximity to cloud artificially enhance the relationship between aerosol loading, cloud albedo, and cloud fraction. These results suggest that previous satellite-based radiative forcing estimates represented in key climate reports may be exaggerated due to including retrieval artefacts in the aerosol located near clouds.

## 1 Introduction

Aerosols are hypothesised to cool the climate system due to their ability to enhance the reflection of clouds (Twomey, 1974; Albrecht, 1989), particularly ubiquitous warm phase clouds located in the boundary layer (Christensen et al., 2016a). The uncertainty attached to this cooling effect is considered to be very large (IPCC, 2013) in both satellite and general circulation model (GCM) estimates. The fundamental issues causing this large uncertainty stem from a myriad of challenges related to retrieval artefacts in satellite products (e.g. cloud contamination and 3D radiative effects) and missing processes in GCMs (e.g. parameterisation schemes related to cloud-top entrainment feedbacks and buffering in aerosol-cloud interactions as discussed in Stevens and Feingold, 2009). Satellite observations suggest that the aerosol indirect radiative cooling effect is about half that of GCM-based estimates (e.g. see Figure 7.19 in chapter 7 of the IPCC 5$^{th}$ Assessment Report IPCC, 2013). For future

climate projections it is critical to close the satellite-GCM gap in the forcing estimate so we can understand to what extent anthropogenic aerosols may have cooled, and continue to cool, the climate system.

Despite the advances in satellite-based retrievals in recent decades, obtaining robust statistical relationships between aerosols and clouds remains difficult. Challenges associated with obtaining accurate passive satellite retrievals generally involve the following issues: (1) an artificially high aerosol optical depth (AOD) retrieval due to the presence of undetected cloud in areas assumed to be cloud-free (cloud contamination) (Remer et al., 2005); (2) radiation scattered by clouds that illuminate the aerosol field and are not accounted for in the 1-D radiative transfer model in satellite retrievals causing erroneously high AOD retrievals (3D effects) (Varnái and Marshak, 2009), and (3) humidification causing aerosols to swell near clouds thereby enhancing AOD without any increase in aerosol number concentration (humidification effect) (Twohy et al., 2009). Attributing which of these mechanisms contribute to the enhancement in AOD and particle size near clouds is difficult to determine alone using passive sensing instruments from satellite-based observations.

Recent assessments characterising near-cloud aerosol retrieval artefacts were examined using spaceborne lidar (Várnai et al., 2013) and ground-based lidar (Ten Hoeve and Augustine, 2016) observations. In general, the MODerate Resolution Imaging Spectroradiometer (MODIS) aerosol products show enhanced aerosol optical depth and particle size several kilometres beyond the cloud edge (Várnai and Marshak, 2015). Evidence from selected Surface Radiation Budget Network (SURFRAD) observations reported in Ten Hoeve and Augustine (2016) suggest that near-cloud contamination in the satellite retrieval can enhance the aerosol cloud cover fraction relationship by approximately 40% (i.e. cloud cover fraction is larger for the same aerosol optical depth when cloud contaminated pixels are used in the analysis). In addition, the direct aerosol radiative effect (i.e. the increase in reflected radiation due to aerosol loading in clear-sky conditions) in satellite observations is significantly larger (35-65%) than that inferred from larger (greater than $20 \text{ km}^2$) cloud-free ocean regions where the aerosol retrievals are located farther from the clouds (Twohy et al., 2009). The extent to which these near-cloud aerosols influence the correlation-based statistics used as diagnostics for aerosol-cloud interactions in satellite observations is largely unknown at the global scale. The quantification of their impacts on the aerosol radiative forcing estimate is the goal of this study.

Besides the errors in the satellite retrieval, attributing causal relationships between non-collocated cloud and aerosol retrievals is challenging. Passive sensors cannot currently retrieve aerosol and cloud simultaneously because the imager pixels are classified as being either "cloud" or "cloud-free." To solve the collocation problem a **pre-averaging** methodology is commonly used (e.g. Sekiguchi et al., 2003; Quaas et al., 2008; Lebsock et al., 2008; Grandey and Stier, 2010; Chen et al., 2014; Gryspeerdt et al., 2016; Christensen et al., 2016a). Typically, the aerosol and cloud properties are pre-averaged over broad regions as a means to encapsulate both retrieval types together within a (level-3 type) grid-box. The aerosol and cloud bins are typically constructed at a spatial scale of $1° \times 1°$ at daily time intervals. This method limits the number of sampling pairs to at most 90 (for a three month period) from which to derive seasonal regression statistics. Some studies have also used a **back-trajectory** model to pair aerosols and clouds together (Bréon et al., 2002) and compute aerosol cloud diagnostics using the assumption that the aerosol properties remain constant over the coarse of the trajectory. A disadvantage to using these approaches is if the artifacts in the pixel-scale retrieved data (i.e. the level-2 data) near clouds is not properly screened

the aerosol-cloud relationships may be biased regardless of the methodology used to combine the observations together from satellite data.

Another, less common method to collocate aerosols and clouds together is **data assimilation**. This process uses aerosol properties that are extracted from reanalysis products at the location of the observed cloud (Bellouin et al., 2013; Amiri-Farahani et al., 2017). However, the aerosols simulated by the models may be strongly affected by wet deposition (Gryspeerdt et al., 2015) particularly in locations where the satellite observations cannot provide a constraint on the aerosol loading in cloudy areas (Christensen et al., 2016a).

In this paper, we have developed a new collocation method, defined here as the high-resolution **Cloud-Aerosol Pairing Algorithm** (CAPA). In this method high-resolution pixel-scale cloud observations are paired to the nearest aerosol retrieval within the satellite image. These pairs are then aggregated over $1° \times 1°$ regions. This results in a significant boost to the total number of samples used in the regression statistics from about 90 (using $1° \times 1°$ pre-averaged daily statistics) to approximately $11 \times 10^5$ using CAPA over a three month period. In addition, the distance between cloud-aerosol retrievals can be adjusted in CAPA as a means to screen for aerosols next to clouds. This method provides the ability to expose biases in the aerosol-cloud radiative forcing estimate due to satellite artifacts in the retrievals of aerosols near-cloud.

The manuscript is organised into the following format: satellite data sets and their corresponding retrieval algorithms are described in section 2, the procedure to compute aerosol indirect radiative forcing in section 3, CAPA is described in section 4, compositing techniques are described in section 5, the results using this approach are compared to the standard pre-averaging method as described in section 6, and the summary and discussion of this work and how it relates to our companion paper (Neubauer et al., 2017) is described in section 7.

## 2   Satellite Data

Two passive sensors with similar equator crossing times (approximately 10:30 am local), the Advanced Along Track Scanning Radiometer (AATSR) on Envisat and MODIS on Terra, are used in this study. AATSR is a dual-view instrument having a footprint resolution of about 1 km at the surface. The two views (satellite zenith angles at $\sim 55°$ forward and nadir ) offered by the instrument provide near-simultaneous (within 90 s) observations of the same location on the Earth. This provides the ability to separate the surface from the atmospheric signals in order to increase the accuracy of the 1-km aerosol retrieval (in version 4.01 the aerosol is reported at 1 km resolution) using the Optimal Retrieval of Aerosol and Cloud (ORAC) algorithm (Thomas et al., 2009). Cloud properties are also retrieved using the same inputs (e.g. the cloud mask, surface reflectance data over the ocean, basis for the optimal estimation scheme in the radiative transfer model, and thermodynamic profiles) as the aerosol retrieval algorithm. By using the same inputs for the ORAC forward model we achieve a high-degree of consistency between the aerosol and cloud products. This consistency is essential to constraining the complexity in attributing cause and effect in aerosol-cloud interaction studies. The ORAC algorithm for cloud is described in Poulsen et al. (2012) and most recent improvements to the forward model code in Sus et al. (2017) and McGarragh et al. (2017). The high-resolution pixel-scale data is currently available through the European Space Agency (ESA) Climate Change Initiative (CCI) although the aerosol products

are averaged to a spatial resolution of 10 km to be consistent with other satellite products of this kind. The pre-averaged gridded daily products (L3C) used in this study are also available via http://cci.esa.int.

Shortwave and longwave broadband radiative fluxes are obtained using the CC4CL (Community Cloud Retrieval for Climate) algorithm (Christensen et al., 2016b). The code uses the BUGSrad radiative transfer model (Stephens et al., 2001) in conjunction with the input cloud and aerosol properties derived from ORAC. BUGSrad is based on the two-stream approximation and correlated-k distribution methods of atmospheric radiative transfer. It is applied to a single-column atmosphere for which the cloud and aerosol layers are assumed to be plane-parallel. Cloud and aerosol properties retrieved using ORAC are ingested into BUGSrad to compute both shortwave and longwave radiative fluxes for the top and bottom of atmosphere. The algorithm uses 18 bands that span the electromagnetic spectrum to compute the broadband flux, 6 in the shortwave and 12 in the longwave. BUGSrad shows good agreement with the Clouds and the Earth's Radiant Energy System (CERES) observations (Henderson et al., 2013) and has been used to assess the Earth's energy budget using CloudSat observations (Stephens et al., 2012).

The MODIS 1-km pixel-scale data is obtained from the Terra satellite. Terra was selected for comparison because it has a similar orbit and equator crossing time as Envisat. To obtain top of atmosphere (TOA) radiative fluxes from MODIS on Terra we ingest the standard collection 6 cloud (MOD06; Platnick et al., 2017) and aerosol (MOD04; Levy et al., 2013) into the CC4CL radiative flux BUGSrad model. Because the MOD04 product is sampled at 10 km spatial resolution the data has been resampled at 1-km resolution to match the cloud product. The ESA CCI projects evaluated several aerosol and cloud products, among them were the standard collection 6 MODIS aerosol and cloud products. ORAC was also applied to MODIS data (within Cloud_cci) and the MODIS-ORAC product was found to agree very well with the MODIS standard products (Hollmann, 2017).

This analysis uses ten years of AATSR observations spanning from 2002 to 2012. Pixels are screened to include only low-level (cloud top pressure greater than 500 hPa) liquid phase (cloud top temperature greater than 273 K) maritime clouds. Due to limitations in data storage we are only able process three months (June, July, and August; JJA) using high-resolution MODIS-ORAC (broadband fluxes derived using standard MODIS products) Terra retrievals for comparison.

## 3  Aerosol Indirect Radiative Forcing Calculation

Computation of the aerosol indirect radiative forcing estimate is based on a top-down approach in which a system-wide variable, the cloud radiative effect (CRE), is used to compute the radiative forcing as a function of the aerosol loading. This reduces the number of free-parameters to just a few (e.g. cloud fraction, cloud albedo, and aerosol index) in which the observational uncertainties are better known (Feingold et al., 2016) compared to other quantities that are difficult to retrieve from passive satellite measurements (e.g. droplet number concentration, cloud condensation nuclei and cloud thickness). The derivation of the shortwave component of the aerosol indirect radiative forcing at the top of the atmosphere is the same as that used in Chen et al. (2014) and is derived from the cloud radiative effect equation written here as

$$F_{\mathrm{CRE}} = F_{\mathrm{clr}} - F_{\mathrm{all\text{-}sky}} \qquad (1)$$

where $F_{clr}$ is the clear sky net radiative flux (i.e., $F_{clr} = F_{clr}^{\uparrow} - F_{clr}^{\downarrow}$, arrows denote the upwelling and downwelling fluxes, respectively) for atmospheric columns in containing no clouds and $F_{all\text{-}sky}$ is the net flux that is observed for all-sky conditions (excluding ice clouds here) computed from clear- and cloudy-sky 1-km pixels located within each $1° \times 1°$ region. The thermodynamic profiles used in the broadband flux calculations are interpolated to each 1-km imager pixel from the N256 spatial resolution of the ECMWF (European Centre for Medium range Weather Forecasting) Interim Reanalysis product. Assuming a relatively dark ocean surface equation (1) can be decomposed into

$$F_{all\text{-}sky} = (1 - c_f)F_{clr} + c_f F_{cld} \qquad (2)$$

where $F_{cld}$ is the component of the radiative flux contributed by clouds and $c_f$ is the cloud cover fraction over the observed area. Combining equations (1) and (2) and considering the shortwave component of the upwelling fluxes the cloud radiative effect becomes

$$F_{CRE} = c_f(A_{clr} - A_{cld})F^{\downarrow} \qquad (3)$$

where $A_{clr}$ is the clear-sky albedo, $A_{cld}$ is the albedo of the cloud and $F^{\downarrow}$ is the daily-mean incoming top of atmosphere solar radiation. Taking the derivative of equation (3) with respect to the aerosol index ($AI$) gives the column TOA cloud radiative effect aerosol sensitivity

$$\frac{dF_{CRE}^{sw}}{d\ln AI} = \left( \overline{c_m}\left[ \frac{dA_{clr}}{d\ln AI} - \frac{dA_{cld}}{d\ln AI} \right] + \overline{(A_{clr} - A_{cld})}\frac{dc_f}{d\ln AI} \right) F^{\downarrow} \qquad (4)$$

where $\overline{c_m}$ is the climatology of clouds having cloud top pressure greater than 500 hPa and composed of liquid phase droplets over ocean regions, the derivative terms represent the change in clear-sky and cloudy-sky albedo as a function of aerosol index. The first term on the right-hand side of equation (4) is called the intrinsic aerosol effect and includes the impact of aerosol on changes in cloud albedo. The second term is the extrinsic effect which represents the impact of aerosol on cloud fraction. For further details regarding the derivation of these terms see Chen et al. (2014).

We use aerosol index ($AI = \tau_a \times Å$, where $\tau_a$ is the aerosol optical depth at 550 nm and $Å$ is the Ångström exponent derived from the optical depths at 550 nm and 869 nm) because $AI$ has been shown to serve as a better indicator of the column cloud condensation nuclei compared to aerosol optical depth (Nakajima et al., 2001; Gryspeerdt et al., 2017). To limit co-variation between the annual cloud and aerosol cycles with the incoming solar radiation flux the aerosol-cloud sensitivities are computed, first, over each season separately and then combined together to form the annual mean aerosol indirect effect sensitivity.

Finally, the aerosol indirect forcing estimate is obtained by multiplying equation (4) by an amount of aerosol attributable to anthropogenic activities as determined by MACC-II reanalysis data (i.e., $\Delta\tau_a^{MACC} = \ln(\tau_a^{MACC}) - \ln(\tau_a^{MACC} - \tau_{anth}^{MACC}) = \ln\left(\frac{1}{1-F_{anth}}\right)$, where $\tau_{anth}^{MACC}$ is the aerosol optical depth attributed to anthropogenic activities). In this calculation we have assumed that the fractional change in the anthropogenic AOD is equivalent to the fractional change in anthropogenic $AI$. This assumption may lead to an underestimation of the aerosol indirect forcing especially in regions where dust dominates (Gryspeerdt et al., 2016). MACC-II is a satellite-model hybrid dataset that utilises the state-of-the-art ECMWF-IFS (Integrated Forecast System) aerosol transport model along with a surface emissions inventory and assimilated MODIS data to provide

aerosol optical depth for a variety of species including dust, organic carbon, sea-salt, black carbon, and sulphate (e.g. see Morcrette et al. (2009) and Benedetti et al. (2009) for details). Global distributions of the anthropogenic aerosol fraction are provided in Chen et al. (2014) and Bellouin et al. (2013); the global oceanic mean value is about 21%.

## 4 Cloud-Aerosol Pairing Algorithm (CAPA)

The basic approach of CAPA is to pair each cloud observation to the nearest aerosol retrieval. This process is performed within the satellite image (the swath width is 2330 km for MODIS and 512 km for AATSR). Distances, using Euclidian geometry in pixel-coordinates, are computed between each cloud pixel and all possible aerosol retrievals. Given the large number of pixels within the swath this task is computationally demanding if proper screening is not carried out initially. To decrease computation time and the number of possible pairs (for a given cloud observation), the satellite image is divided into sections and pixels

are grouped together (typically $250 \times 250$ pixel regions). For pixels near the edge of each square region (within 125 km) the search radius is extended into the nearest adjacent square region. These steps eliminate aerosol-cloud pairs at distances greater than approximately 150 km (for 1-km pixel-scale data), but potential pairs beyond this length-scale occur at low frequencies and are probably less relevant since the cloud-aerosol interaction is less likely to be influenced by the same airmass (Anderson et al., 2003).

In the next step the cloud fraction is computed (using the satellite retrieval cloud mask at 1-km resolution) as a metric to determine the appropriate search algorithm to use for the given subsection of the satellite granule. If the cloud fraction is high ($c_f > 50\%$) then fewer aerosol retrievals exist and the distance between each cloud pixel and all aerosol pixels is computed simultaneously via brute force within that region of the satellite orbit. However, if the cloud fraction is low ($c_f < 50\%$) the number of aerosol targets will be larger and the former approach will run much slower. In this case the pixels that are adjacent

to the cloud observation are searched first (as there is a relatively high probability an aerosol is located in an adjacent pixel when the cloud fraction is lower). If an aerosol retrieval is not found in the adjacent pixels this step is repeated, again, until an aerosol target is found. By using both search algorithms together (based on cloud fraction) the computation speed decreases by more than 50% compared to running the retrieval in brute-force mode alone. Finally, if two (or more) aerosol pixels are located at the same distance from the cloud observation then one of them is selected at random. The CAPA run time is approximately

2-6 minutes using a single core Intel/AMD at $\sim 3$ GHz processor for a typical MODIS granule that contains 2.5 million 1-km pixels.

The results of CAPA are applied to part of an AATSR orbit and displayed in Figure 1. The image shows a belt of clouds across the middle of the granule with aerosols retrieved on both sides (Figure 1b). The cloud-belt is approximately 300 km wide (Figure 1c). Aerosols in the lower half of the image have lower optical depths and tend to be located at greater distances to

the nearest cloud (green pixels), whereas the aerosols in the upper half of the image are retrieved mostly in a broken cloud field in which the aerosols are located close to the clouds (red pixels). The aerosol retrievals have a "blocky" appearance because the standard level-2 aerosol products from MODIS (MOD04) and AATSR (V4.02) are averaged over larger 10 km$^2$ pixels. In

order to match the cloud and aerosol products together and form aerosol-cloud pairs (on the same imager pixel grid) the aerosol products for AATSR and MODIS are resampled at 1 km resolution.

Aerosol statistics are examined as a function of distance to the nearest cloud in Figure 2. The observations are comprised of three months (JJA in 2008) of data collected across four different $10° \times 10°$ regions (off the coasts of California, Peru, Azores, and Namibia). In every region, the mean aerosol optical depth increases, the Ångström exponent decreases and the aerosol index increases as the observations are closer in distance to the nearest cloud. In agreement with previous studies we find that about half of all clear-sky columns occur within $4 - 5$ km of the low-level cloud (Várnai and Marshak, 2012). As a consequence, near-cloud aerosols (that are potentially affected by retrieval artefacts) in pre-averaged aerosol datasets would provide substantial weight on the aerosol statistics.

Despite efforts to limit contamination in standard MODIS collection 6 retrievals - threshold values for the darkest and brightest 25% of pixels in each cluster (Remer et al., 2005) are applied. Although one complication with the standard MODIS products is the cloud and aerosol products use different cloud masks to perform their retrievals. Nevertheless, large aerosol optical depths remain in the MODIS observed pixels near cloud edges, due primarily to 3D effects (Varnái and Marshak, 2009) and swelling of aerosols by higher relative humidity. Currently, no effort is made to remove near-cloud pixels from the AATSR ORAC gridded-products. This is evident from the faster and less pronounced decrease in aerosol optical depth and Ångström exponent with distance from the nearest cloud in the MODIS product. At distances greater than 15 km the aerosol optical properties tend to be fairly constant. Similarly, Varnái and Marshak (2009) also noted that beyond 15 km contamination effects were minimised in MODIS data. Therefore aerosol-cloud relationships are also examined for pairs beyond the 15 km length-scale.

## 5   Data Stratification

Results are based on four distinct composites of the data (listed in Table 1). Each composite contains aerosol and cloud pairs which are aggregated into $1° \times 1°$ regions so that regression statistics can be computed on the basis of the linear least-squares slope between the cloud albedo and the aerosol index. This process is also carried out separately for pixels designated as "cloud-free" in order to derive the clear-sky albedo change as a function of aerosol loading (i.e. $\frac{dA_{\text{clr}}}{d\ln AI}$) in equation (4). Two distinct methodological frameworks are used, 1) the high-resolution CAPA and 2) the low-resolution pre-averaging methods.

CAPA is run using two distinct length-scales: 1) clouds are paired with the nearest located aerosol (CAPA-L2) and with the nearest aerosol that is located at least 15 km away from any other cloud (CAPA-L2_15km). The median distance between aerosol and cloud pairs from these two composites is 8.2 and 27.1 km, respectively thereby providing the ability to screen for aerosol next to cloud. Another advantage of CAPA is that it retains individual L2 pixels, i.e. full-resolution data, for quantifying aerosol-cloud relationships over each $1°$ grid-box allowing for more of the aerosol field to be sampled, thus providing more degrees of freedom which to derive seasonal statistics. For example, the average number of high-resolution samples going into a typical $1°$ region over a three month period is approximately $3.0 \times 10^5$, although the actual number of degrees of freedom is much smaller (each region provides on average about 10 degrees of freedom per day providing roughly 1000 over a three

month period) owing to oversampling the same aerosol used to make cloud-aerosol pairs. Nevertheless, the typical number of samples in the CAPA composites far exceed those that can be achieved using the standard pre-averaged products (PRE_AVG-L3 and PRE_AVG-L3_Corr.) where at most they provide 90 samples (providing one aerosol-cloud pair sample per-day for a given grid-box) in a three month period. As pointed out later, the nearly one full order of magnitude fewer unique aerosol-cloud

pairs in the PRE_AVG-L3 composite results in a larger 1-$\sigma$ standard error regression error estimate.

## 6   Results

Aerosol-cloud relationships are examined using the CAPA and pre-average methods at a variety of spatiotemporal scales. An assessment of CAPA is carried out at both the regional-scale ($10° \times 10°$) in selected locations with predominately low-level clouds (i.e. off the coasts of California, Peru, Azores, and Namibia) and at the global-scale.

### 6.1   Global distributions

The intrinsic aerosol-cloud radiative effect is predominately influenced by the change in cloud albedo as a function of the aerosol index (i.e. $\frac{dA_{cld}}{d\ln AI}$). The regression is quantified using a range of temporal averaging periods using AATSR observations in Figure 3. As more years of observations are included in the regression the global distribution becomes less noisy (as represented by a decrease in the standard deviation of the regional-scale spatial distribution from $\sigma = 0.05$ to $0.02$ for the

PRE_AVG-L3 composite). Although, beyond 5 years the differences in the intrinsic effect between consecutive years become statistically insignificant. We conclude that 10 years of AATSR observations are more than adequate to construct these diagnostics. Regarding MODIS, we acknowledge that the limited 3-month time period may bias the standard error of the regression slope. Therefore, it is used primarily to test the new CAPA method.

Figure 4 shows the global distribution of the observed intrinsic aerosol indirect radiative forcing estimate using AATSR

observations. Consistent with Chen et al. (2014), larger values are observed in the Northern Hemisphere in accordance with the larger fractions of anthropogenic aerosol and maritime low-level clouds. Another, more notable, result displayed in Figures 3 & 4 is the substantial decrease in the strength of the forcing when using the CAPA algorithm to screen for aerosols in the vicinity of clouds (i.e. between CAPA-L2_15km and the CAPA-L2 and PRE_AVG-L3 composites). The mean forcing estimate is smaller partly because there are a larger fraction of grid-boxes with negative cloud albedo sensitivities from just 11% (in

CAPA-L2) to 31% (in CAPA-L2_15km). Because all three composites use the same cloud fraction and anthropogenic aerosol fraction climatologies the radiative forcing differences are primarily due to the cloud albedo sensitivities as inferred from equation (4) and displayed in Figure 3.

### 6.2   Regression Tests at Regional-Scales

To understand why the cloud albedo effect sensitivity is weaker when aerosols are removed close to clouds we examine the

diagnostics in several regions across the globe. Figure 5 shows the mean cloud albedo plotted as a function of the aerosol index for the California region. The linear regression slope between cloud albedo and $AI$ is steeper when the clouds are paired to

contaminated aerosols located within 15 km from another cloud. The cloud albedo also tends to increase monotonically as a function of $AI$ until reaching a value of about 0.25 in the CAPA-L2 composite. Beyond this value cloud albedo increases at a slower rate because the clouds become less susceptible in a more polluted atmosphere. By contrast, when pairing the same clouds to aerosols that are located at least 15 km from another cloud the aerosol index values shift to smaller values and the slope between cloud albedo and $AI$ decreases. Similar responses are also found in thicker clouds that are less "susceptible" (Platnick and Twomey, 1994) to aerosol perturbations (red points in Figure 5 and Figure 6) as well as in three other regions (Figure 6). These results suggest that aerosols located in the vicinity of clouds may be more likely to contaminate and inflate the statistical relationships between cloud and aerosol properties.

### 6.3 Aerosol-Cloud Radiative Forcing Estimates

Intrinsic aerosol-cloud radiative forcing estimates are provided for each composite in Figure 7. The estimates from AATSR and MODIS agree with each other and with the reported value from the MODIS Aqua and CERES afternoon-train observations in Chen et al. (2014) to within 0.1 W m$^{-2}$. This relatively good agreement occurs despite the use of different temporal averaging periods. Furthermore, it is found that when no aerosol screening takes place in the pre-averaged gridded (PRE_AVG-L3) and high-resolution cloud-paring data sets (CAPA-L2) the forcing estimates are nearly two-times larger than the composites that screen for aerosol near cloud. This is due to the weaker relationship between cloud albedo and $AI$ sensitivity for the aerosols selected farther away from clouds (CAPA-L2_15km).

Extrinsic aerosol-cloud radiative forcing estimates are shown in Figures 8 and 9. In agreement with previous studies (e.g. Quaas et al., 2010; Chand et al., 2012; Grandey et al., 2013; Gryspeerdt et al., 2014) we find the $c_f$-$AI$ relationship is strongly positive in most locations thereby producing a very strong radiative forcing (Figure 8). However, this affect can be corrected. After reconstructing the pre-averaged aerosol data set based on screening out the aerosols close to clouds (PRE_AVG_Corr. composite) this process results in a decreased extrinsic aerosol indirect forcing estimate by approximately 70%. This result agrees with the ground-based SURFRAD observations reported in Ten Hoeve and Augustine (2016) in which the removal of aerosols in the vicinity of clouds decreased the strength of the cloud fraction aerosol loading relationship. Furthermore, because cloud fraction co-varies with relative humidity this estimate is still likely to be over estimated and could be further mitigated through the use of cloud droplet number concentration (Gryspeerdt et al., 2016).

Since numerous studies have used pre-averaged level-3 type aerosol products we have tested whether these products can be corrected. Here, we have reconstructed the pre-averaged aerosol product by removing near-cloud aerosols in the standard AATSR and MODIS data. The aerosol pixel is removed if any part of the $10 \times 10$ km$^2$ area spanning the aerosol sample is located within 15 km from the nearest cloud. In general, similar forcing estimates are obtained when the cloud-contaminated aerosol is screened in the pre-averaged $1° \times 1°$ products compared to the high-resolution aerosol-screened CAPA-L2_15km data.

On average the CAPA methods produce smaller 1-$\sigma$ standard error regression uncertainties (by up to 15%) due to including a larger number of unique cloud-aerosol sampling pairs over each $1°$ region. It is noteworthy that the actual number of degrees of freedom may be somewhat smaller, however, due to high spatial autocorrelation between sampled aerosol in each region as

identified in many studies (e.g. Anderson et al., 2003; Schutgens et al., 2016; Kovacs, 2006; Santese et al., 2007; Shinozuka and Redemann, 2011). Nevertheless, the strength of the aerosol indirect forcing estimate is similar between the corrected pre-averaged products and CAPA if strict screening of near-cloud aerosols are carried out first.

Overall, the intrinsic radiative forcing estimate of $-0.49 \pm 0.18$ W/m$^2$ agrees with the values reported in previous satellite-based studies (e.g. Sekiguchi et al., 2003; Quaas et al., 2008; Lebsock et al., 2008; Bellouin et al., 2013; Chen et al., 2014; Christensen et al., 2016a). However, when the aerosols are removed from the vicinity of clouds these methods produce radiative forcing estimates that are smaller by about 40% giving a new value of $-0.28 \pm 0.26$ W/m$^2$. In addition, the extrinsic forcing decreases by 70% (from $-0.60 \pm 0.24$ to $-0.24 \pm 0.40$ W/m$^2$) which is similar in strength to the radiative forcing metrics used to establish cloud fraction changes to increases in aerosol reported in Gryspeerdt et al. (2016). These results suggest that satellite-based estimates of the effective radiative forcing due to aerosol-cloud interactions represented in key climate reports (e.g. see the IPCC, 2013) may be exaggerated due to retrieval artefacts in the aerosol properties next to clouds. A summary of the estimates derived for oceanic regions are reported in Table 2.

## 6.4 Uncertainty Analysis

We have assumed in this study that the aerosols are fairly homogenous across large spatial scales, up to 150 km according to the results presented in numerous studies examining the spatial autocorrelation length-scale of aerosol optical thickness (e.g. Anderson et al., 2003; Schutgens et al., 2016; Kovacs, 2006; Santese et al., 2007; Shinozuka and Redemann, 2011). However, further analysis has been carried out here to address the spatial scale dependence of the distance between the aerosol and cloud data. Using the observations from AATSR we run an additional test in which the aerosol is removed from nearby clouds up to a distance of 30 km and then each cloud is paired to the nearest far-field aerosol pixel at this scale. Overall, the aerosol indirect forcing estimate is somewhat smaller in strength using 30-km scaling ($-0.20 \pm 0.26$ W/m$^2$) compared to the scaling at 15 km ($-0.28 \pm 0.27$ W/m$^2$) but the differences between the composites are insignificant. Therefore, this suggests that the far-cloud aerosol statistics are representative of the same airmass as those found closer to clouds.

Besides the intrinsic/extrinsic radiative effect concept we apply the CAPA dataset to the aerosol indirect effect method developed in Quaas et al. (2008). The two methods have already been tested against each other and shown to agree very well for satellite observations in the North Atlantic (Amiri-Farahani et al., 2017). A fundamental difference between them are the additional susceptibility terms introduced by the Quaas et al. (2008) method: the first aerosol indirect effect ($\Delta F^{AIE} = \overline{c_{\mathrm{m}}} \cdot A(c_{\mathrm{m}}, \tau_c) \frac{1}{3} \cdot \frac{d \ln N_{\mathrm{d}}}{d \ln AI} F^{\downarrow} \Delta \tau_{\mathrm{a}}^{\mathrm{MACC}}$, where the cloud droplet concentration, $N_{\mathrm{d}}$, is computed from cloud optical thickness and cloud droplet effective radius retrieval assuming an adiabatic approximation for clouds, $A(c_{\mathrm{m}}, \tau_c)$ is given in the Appendix of Quaas et al. (2008), and $\tau_c$ is the cloud optical thickness) that includes the sensitivity of the planetary albedo (the planetary albedo is obtained from the fitting parameters used in Quaas et al., 2008) to a relative change in the cloud droplet number concentration and the sensitivity of cloud droplet number concentration to aerosol index ($\frac{d \ln N_{\mathrm{d}}}{d \ln AI}$). Also, this formula contains a remainder term ($\Delta F^{AIE2} = [(\alpha - (a_1 + a_2 \ln AI)) + \overline{c_{\mathrm{m}}} \cdot A(c_{\mathrm{m}}, \tau_c) \cdot (\frac{d \ln c_{\mathrm{m}}}{d \ln AI} + \frac{d \ln L}{d \ln AI})] F^{\downarrow} \Delta \tau_{\mathrm{a}}^{\mathrm{MACC}}$, where $a_1 - a_6$ are the fitting parameters and $\alpha$ is the planetary albedo) that may correspond to the cloud lifetime effect and includes log changes in both liquid water path ($L$) and cloud fraction as a function of $AI$ (where we have replaced the aerosol optical depth in the

original formulation with $AI$). In general, the total forcing estimate (i.e. adding the intrinsic and extrinsic terms together) tends to agree very well using the Quaas et al. (2008) approach (i.e. adding the first and second indirect effects together) but with the exception of the somewhat smaller PRE_AVG composite results. Using AATSR-ORAC data the PRE_AVG ($-1.1 \pm 0.8$ W/m$^2$) and CAPA-L2 ($-0.5 \pm 0.3$ W/m$^2$) composites are significantly larger than the PRE_AVG_Corr. ($-0.49 \pm 0.75$ W/m$^2$)

and CAPA-L2_15km ($-0.26 \pm 0.25$ W/m$^2$) composites using the Quaas et al. (2008) statistical method. Furthermore, the first aerosol indirect forcing estimates for the PRE_AVG ($-0.14 \pm 0.15$ W/m$^2$) and CAPA-L2 ($-0.13 \pm 0.11$ W/m$^2$) composites are also larger than the PRE_AVG_Corr. ($-0.09 \pm 0.16$ W/m$^2$) and CAPA-L2_15km ($-0.11 \pm 0.14$ W/m$^2$) but these differences (across composites) are less pronounced compared to using the total forcing estimates. Nonetheless, these methods provide supporting evidence that aerosols located closer to clouds enhance the aerosol-cloud radiative effect.

## 7 Conclusions

Two independent satellite instruments, AATSR on Envisat and MODIS on Terra, were used in this study to test a new cloud-aerosol pairing algorithm to compute aerosol indirect forcing estimates for warm oceanic maritime clouds. Cloud-aerosol pairs formed at the pixel-scale resolution of the satellite imager using CAPA can provide a larger sample size from which to compute correlation statistics between cloud albedo and aerosol index. The effect of the larger sample size effectively decreases

the standard error of the regression slope thereby providing higher confidence in the new radiative forcing estimates.

The scheme was also developed to determine the extent to which artefacts in aerosol retrievals located in the vicinity of clouds affect aerosol-cloud relationships in spaceborne satellite observations. By removing aerosols located within 15 km of the nearest cloud, both the cloud albedo effect (intrinsic) and cloud fraction effect (extrinsic) forcing decreased by 40% and 70%, respectively. These new estimates suggest that aerosol effects on the radiative properties of clouds are even smaller than

previously demonstrated from satellite-based studies. This new methodology therefore further widens the gap between satellite and the very strong forcing estimates derived using most GCMs.

One inherent limitation to CAPA is it cannot be used to estimate the extrinsic (or overall) aerosol indirect forcing at the pixel-scale resolution of the satellite imager (typically at 1 km). This is because the pairing algorithm uses pixels that are cloudy to do the pairing thereby resulting in a grid-box mean cloud fraction value of 1.0. It is also generally not practical to use

level-2 pixel-scale data in satellite-based assessments due to the large volume of data that is required at this scale. Therefore, we propose using CAPA (or another analog) to remove the potentially contaminated aerosols within 15 km from nearby clouds as a first step to constructing pre-averaged level-3 type datasets. We demonstrate using 10 years of AATSR data that the broader-scale level-3 data can be corrected to yield the same intrinsic aerosol-cloud sensitivities using the level-2 pixel-scale thereby providing a means to also compute the extrinsic aerosol indirect radiative forcing using the PRE_AVG broader-scale level-3

data.

This initial version of the CAPA algorithm has been used here to highlight a potential source of error in satellite-based estimates of aerosol indirect forcing. The CAPA method highlights areas where we have trust in the satellite-based retrievals that are pertinent for aerosol-cloud interactions. In subsequent versions of this algorithm additional steps could be pursued to

increase computational efficiency. There are more efficient algorithms for finding nearest neighbours in a large dataset. Binary search trees, such as a k-dimensional or vantage-point trees would probably work very well within CAPA. Furthermore, this work would benefit from a deeper examination of the coupling between aerosol-cloud pairs using a back trajectory model (such as the Hybrid Single-Particle Lagrangian Integrated Trajectory) following the method described in Bréon et al. (2002).

5     Extension of this method in comparison with GCMs is explored in the companion paper of Neubauer et al. (2017). Furthermore, this companion paper also quantifies the impacts of meteorology on the aerosol-cloud relationships using numerous meteorological regimes based on lower tropospheric stability and free tropospheric relative humidity. These regimes are used for comparison with the global aerosol-climate model ECHAM6-HAM2. Also, the new CAPA diagnostics are used to select aerosols that are located far from clouds thereby providing a more consistent comparison to the GCM which simulates interactions using dry-mode aerosols. The influence of dry-mode aerosols are also found to produce a much smaller indirect forcing estimate due to the dry aerosol being a better proxy for cloud condensation nuclei. These aspects are developed and explored further in Neubauer et al. (2017).

## 8   Code availability

Code to process aerosol, cloud, and broadband fluxes using ORAC can be obtained via https://github.com/ORAC_CC/ORAC.

## 9   Data availability

The Centre for Environmental Data Analysis (CEDA; http://www.ceda.ac.uk) provided the AATSR satellite data. NASA Goddard (https://ladsweb.nascom.nasa.gov) provided the MODIS satellite data used in this paper.

*Author contributions.* Matthew Christensen developed the CAPA algorithm and applied it to the satellite data sets for the analysis in this paper. David Neubauer had the original idea for and helped in the design of the CAPA algorithm. Caroline Poulsen produced the AATSR-ORAC cloud products, Gareth Thomas produced the AATSR-ORAC aerosol products. Greg McGarragh, Adam Povey, Simon Proud and Don Grainger provided support needed to run ORAC. Matthew Christensen wrote the paper with comments from all co-authors.

*Competing interests.* The authors declare that they have no conflict of interest

*Acknowledgements.* We would like to thank our international members of the ORAC team at Deutscher Wetterdienst, for all of the support in running the retrieval algorithm. We would also like to thank Johannes Quaas for providing the planetary albedo data. CEDA provided the computational infrastructure of JASMIN-CEMS needed to process this data. This research was completed as part of ERACE (The Environmental Response to Aerosols Observed in CCI ECVs), being a program of, and funded by, the European Space Agency through a Living Planet Fellowship and the Cloud_cci (contract: 4000109870/13/I-NB) and the Aerosol_cci projects (ESA Contract No. 4000109874/14/I-NB). This study was also funded as part of NERC's support of the National Centre for Earth Observation.

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

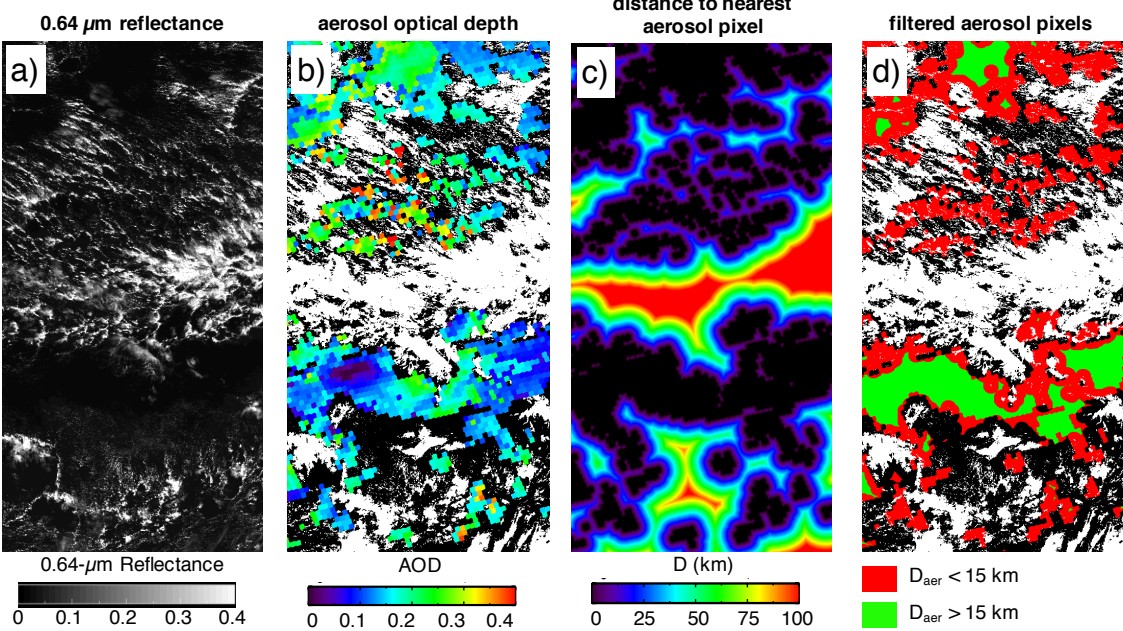

**Figure 1.** a) Satellite image of the visible reflectance at 0.64 μm from part of an AATSR orbit (512 ×1000 km²) on the 20 June 2008 over the ocean off the west coast of Africa. b) Coinciding cloud mask (white) and aerosol optical depth (rainbow) retrieved using ORAC and c) distance from each pixel to the nearest aerosol retrieval. d) Aerosol retrievals that are located within 15 km of a cloud are considered contaminated (red) while those farther away than this are considered valid (green).

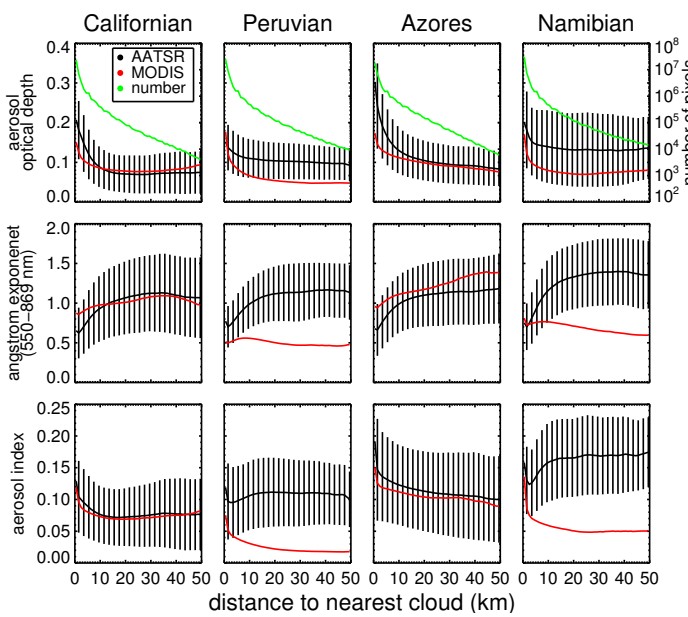

**Figure 2.** Mean aerosol optical depth as a function of the distance to the nearest cloud mask averaged over 1-km width bins. Relationships are plotted using AATSR-ORAC (black) and collection 6 MOD04 applied to MODIS-ORAC (red) for JJA 2008 data grouped into $10° \times 10°$ regions off the coasts of California ($20° - 30°$N, $140° - 130°$W), Peru ($10° - 20°$S, $80° - 90°$W), Azores ($15° - 25°$N, $25° - 35°$W), and Namibia ($10° - 20°$S, $0° - 10°$E). The number of AATSR pixels in each bin are plotted over the results (green). Error bars are denoted by the 1 standard deviation computed over the collection of AATSR retrievals within the bin.

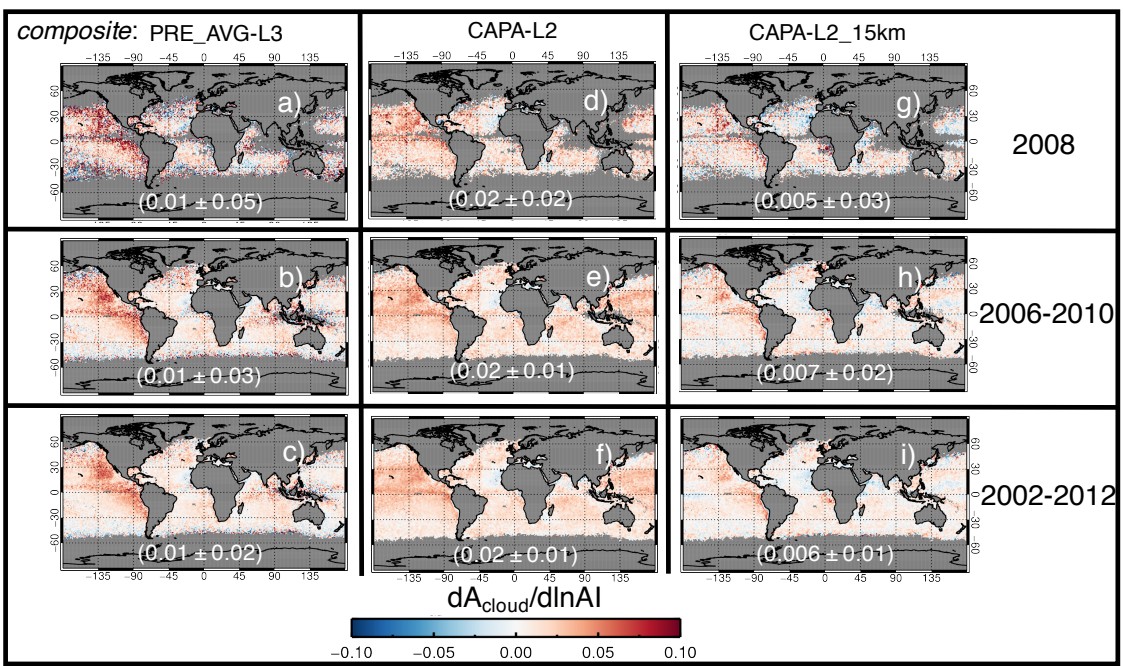

**Figure 3.** Cloud albedo sensitivity to changes in aerosol index averaged over $1° \times 1°$ regions using AATSR. Columns correspond to composites of the data based on the method selection in which the data is pre-averaged (a,b,c) using level 3 daily gridded regions (left column; Level 3), paired to the nearest high-resolution aerosol retrieval (d,e,f), and paired to nearest high-resolution aerosol retrieval that is at least 15 km away from a cloud (g,h,i). Paired observations are grouped into JJA periods using 1-year for 2008 (a,d,g), 5-years from 2006-2010, and 10 years from 2002-2012. Mean values and standard deviations of the spatial distribution are provided in parenthesis for each plot.

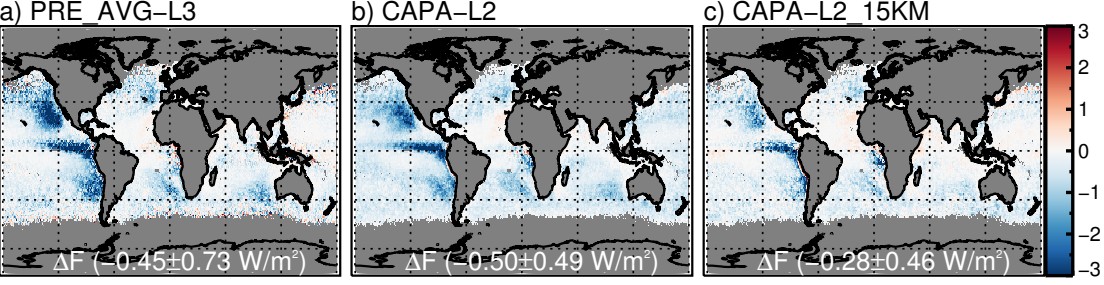

**Figure 4.** Intrinsic aerosol indirect radiative forcing computed using AATSR observations over oceanic $1° \times 1°$ regions during the period $2002 - 2012$. Oceanic mean and standard deviation from the spatial distribution of the forcing ($\Delta F$) is given in parenthesis.

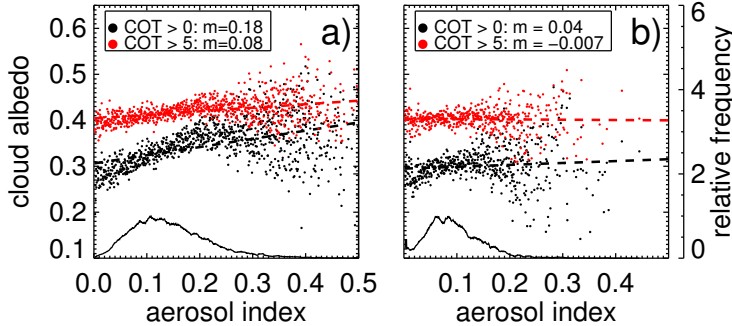

**Figure 5.** Binned cloud albedo as a function of aerosol index based on high-resolution aerosols that are paired to a) the nearest cloud observation (CAPA-L2 method) and b) the nearest cloud observation that is at least 15 km away (CAPA-L2_15km method) over JJA 2008. Cases are binned by 0.001 wide bins in $AI$ over the California region ($20° - 30°$N, $140° - 130°$W) and distribution of the relative frequencies over each bin for $AI$ are plotted over the results. Both methods are composited further by selecting cloud retrievals with cloud optical thickness (COT) greater than 5 (red points). Relative frequency of the number of cloud-aerosol occurrence is plotted beneath the scatter plot. Least squares fit line and value of the slope is provided for each composite.

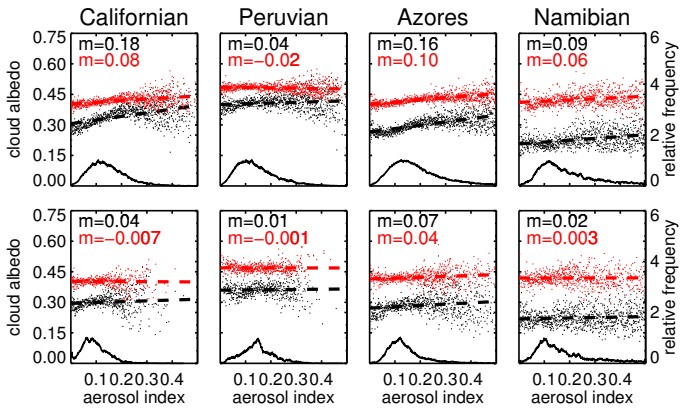

**Figure 6.** Same as Figure 5 but for additional regions including Peru ($10° - 20°$S, $80° - 90°$W), Azores ($15° - 25°$N, $25° - 35°$W), and Namibia ($10° - 20°$S, $0° - 10°$E).

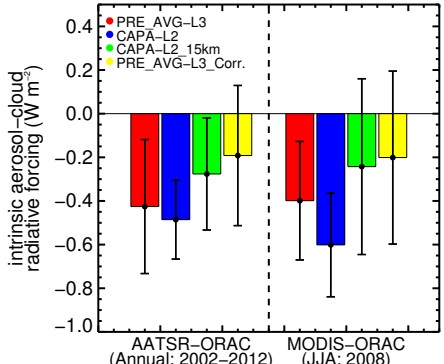

**Figure 7.** Estimated intrinsic aerosol-cloud radiative forcing (as defined in equation 4) by global marine warm clouds derived using top of atmosphere shortwave fluxes from AATSR-ORAC (left columns) and MODIS-ORAC (right columns). Estimates are provided using the pre-average standard level-3 products (PRE_AVG-L3 composite; red), and pre-average corrected level-3 products in which the nearest aerosol is located at least 15 km from any other cloud (PRE_AVG-L3_Corr.; yellow), cloud pairs using the nearest high-resolution aerosol retrieval (CAPA-L2; blue), cloud pairs using the nearest high-resolution aerosol retrieval that is at least 15 km away from any other cloud (CAPA-L2_15km; green). Error bars are calculated on the basis of the standard error of the regression slope.

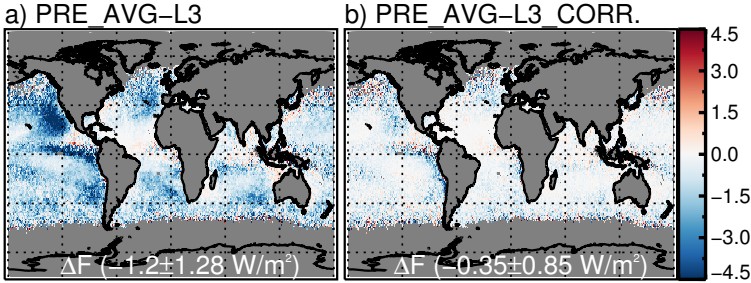

**Figure 8.** Extrinsic aerosol indirect radiative forcing computed using AATSR observations over oceanic $1° \times 1°$ regions during the period $2002 - 2012$. Oceanic mean and standard deviation of the forcing ($\Delta F$) is given in parenthesis.

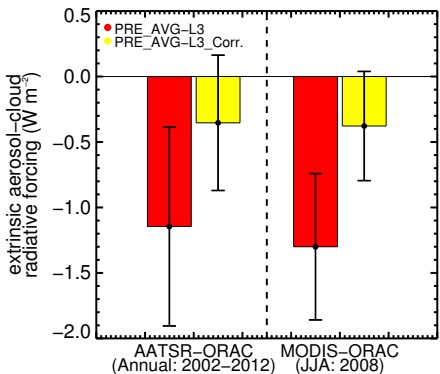

**Figure 9.** Estimated extrinsic aerosol-cloud radiative forcing (as defined in equation 4) by global marine warm clouds derived using top of atmosphere shortwave fluxes from AATSR-ORAC (left columns) and MODIS-ORAC (right columns). Estimates are provided using the pre-average standard level-3 products (PRE_AVG-L3 composite; red), and pre-average corrected level-3 products in which the nearest aerosol is located at least 15 km from any other cloud (PRE_AVG-L3_Corr.; yellow). Error bars are calculated on the basis of the standard error of the regression slope.

**Table 1.** Summary of composites used in this study

| Composite | Description |
|---|---|
| Nearest high-resolution cloud-aerosol pair (CAPA-L2) | Comprised of level-2 individual pairs of pixels in which the length-scale between the cloud and nearest high-resolution aerosol retrieval can range from 0 to 150 km. An example of the pixel selection for these cases is displayed in Figure 1d; red and green pixels. |
| Nearest high-resolution cloud-aerosol pair separated by at least 15 km (CAPA-L2_15km) | Same as above except the nearest high-resolution aerosol retrieval has to be located at least 15 km from any other cloud and cannot exceed a pairing length-scale beyond 150 km. An example of the pixel selection for these cases is displayed in Figure 1d; red pixels. |
| Pre-averaged cloud & aerosol (PRE_AVG-L3) | Pairs are based on un-colocated level-3 pre-averaged $1° \times 1°$ cloud and aerosol observations from standard MODIS (i.e. MOD08) and AATSR products. |
| Pre-averaged cloud & corrected aerosol (PRE_AVG-L3_Corr.) | Same as above except the cloud observations are paired to a new aerosol dataset based on pre-averaging only those aerosol pixels that are located at least 15 km away from cloud. |

**Table 2.** Aerosol-cloud radiative forcing estimated using equation 4 based on ORAC applied to AATSR over the $2002 - 2012$ period and ORAC applied to MODIS collection 6 over JJA-2008 for low-level warm maritime clouds observed between $60°S - 60°N$. Retreivals are not used if they are identified over land or ocean regions with sea ice. Uncertainties are calculated on the basis of the propagated standard error of the regression slope through the radiative forcing calculation.

| | Intrinsic Forcing (W m$^{-2}$) | | Extrinsic Forcing (W m$^{-2}$) | |
|---|---|---|---|---|
| | AATSR | MODIS | AATSR | MODIS |
| Nearest high-resolution cloud-aerosol pairs (CAPA-L2) | $-0.49 \pm 0.18$ | $-0.60 \pm 0.24$ | N/A | N/A |
| Nearest high-resolution cloud-aerosol pairs aerosol is at least 15 km from any cloud (CAPA-L2_15km) | $-0.28 \pm 0.26$ | $-0.24 \pm 0.40$ | N/A | N/A |
| Pre-averaged 1° cloud & aerosol pairs (PRE_AVG-L3) | $-0.43 \pm 0.31$ | $-0.40 \pm 0.27$ | $-1.1 \pm 0.76$ | $-1.3 \pm 0.56$ |
| Pre-averaged 1° cloud & corrected aerosol pairs pre-averaged aerosols are at least 15 km from clouds (PRE_AVG-L3_Corr.) | $-0.19 \pm 0.32$ | $-0.20 \pm 0.40$ | $-0.35 \pm 0.52$ | $-0.38 \pm 0.42$ |