# Peer review of "Unveiling aerosol-cloud interactions Part 1: Cloud contamination in satellite products enhances the aerosol indirect forcing estimate"

_Atmospheric Chemistry and Physics, 2017_

## Referee Comment (RC1) · Anonymous Referee #1 · 21 Jun 2017

Christensen et al. present a new technique of relating aerosol- and cloud retrievals from satellite data. They created an algorithm that searches for the nearest aerosol retrieval for each cloud retrieval. Different from previous approaches (Bréon et al., 2002), no backtrajectories are computed, but the nearest pixel, independent on whether or not the aerosol might actually be advected to the cloudy region. Despite this, it is an innovative approach and may indeed help overcome some issues with the approach still commonly used to relate 1°x1° average aerosol- and cloud retrievals. The authors analyse statistical relationships between the aerosol index and cloud albedo computed on the basis of satellite cloud retrievals using a radiative transfer code, as well as between AI and cloud fraction. They proceed to compute implied radiative forcings.

The manuscript is astonishingly superficial in many of the explanations. Many statements are very difficult to follow, or not at all reproduceable from the information provided.

The authors are imprecise in their language. It seems they in general want to assess the effective radiative forcing due to aerosol cloud interactions, i.e. the overall cloud response to the aerosol, including cloud water path and cloud fraction changes.

Nevertheless, it is a useful paper and should eventually be published. However, I have numerous specific points the authors should address.

P1 L17 Not so much in satellite estimates

L20 the "buffering" isn't precisely defined. A better more specific explanation on what is missing is necessary

P2 L1 given the large range of GCM estimates, it needs to be clarified which publications the authors refer to

L4 remains

L4-10 the order is awkward. If one had proper CCN retrievals (in the order the authors impose item 5), items 1-3, perhaps even 4, wouldn't matter. Also not all problems are pertinent to all aerosol-cloud interactions. The authors need to be specific about what exactly they want to study and where which of the issues arises.

L11 the authors need to clarify what they mean by "contamination" (do they mean problem 1, 2, or 3?) 3, to some extent 2, cannot really be called "contamination" since these are plausible physical processes. It is also important that the authors shouldn't forget to mention that clouds are also an actual source of aerosol. Sulfate predominantly nucleates via the aqueous phase.

L19 "larger" than what? And do the authors really refer to a forcing here, or rather to an effect?

L29 that hold of course only if one analyses one grid box over one season. Experience shows that in such attempts, very rarely 90 data points would be available.

L31 of course also the problem of spurious clouds in pixels labelled cloud -free

P3 L1 this statement needs further explanation to be understandable.

L3 While the authors call their method "new" they should acknowledge at the presumably first aerosol-cloud interaction study from satellites (Bréon et al., 2002, doi:10.1126/science.1066434) already applied such a method.

L7 the theoretical maximum for the 1km MODIS retrievals of clouds is about 110x110x90 = 11x10ˆ5. Is the reduction by a factor of 3-4 an empirical result?

L8 Can the authors clarify what the scale of the MODIS retrievals is? I believe it is 1 km for the cloud product, but is it also 1 km for the aerosol product?

L16 It would be good to report the overpass time

L18 "seconds" should be abbreviated ("s")

l24 the authors should explain their statement "this consistency is essential". The conclusion is not straightforward, but obviously using the same cloud mask for aerosol- and cloud retrievals also introduces issues.

L26 bracket awkward

p4 l7 The appropriate reference for MODIS collection 6 cloud products is Platnick et al. (2017, doi:10.1109/TGRS.2016.2610522)

l22 "lower" than what?

L27: $\F^\downarrow_\mathrm{clr}$: why the index "clr", this is just the incident solar radiation, it seems? Why not operate in Eq. 1 simply with reflected fluxes that are actually observed? Also, at a pixel level, CRE is not defined from observations (cloud fraction is either one or zero). At which scale do the authors compute the CRE?

L28: It is a bit misleading to call $F_\mathrm{obs}$ "observed". It obviously rather is the flux computed on the basis of the aerosol and cloud retrievals. Why not "all-sky" as usually defined? Are the clear-sky thermodynamic profiles from reanalysis for the appropriate grid cell? Is the humidity for these the all-sky or the clear-sky humidity? In which sense is "clr" less "observed" than "obs"? Isn't that applying the retrieved aerosols?

P5 l4 This is a very loose definition of an "indirect effect". The authors can of course define such a quantify. Usually one would call the definition in Eq. 4 something like a "cloud radiative effect sensitivity", and if one multiplies this with the anthropogenic \Delta AI, one would obtain a proxy for the effective forcing due to aerosol-cloud interactions (proxy since it only accounts for column physics).

L8 it is peculiar that the authors make use here of the annual-mean incoming solar radiation. The factor in the brackets presumably has a strong diurnal and annual cycle. Co-variation of this factor with the incoming solar radiation then leads to possibly substantial differences in the radiative effects compared to the ones proposed by the authors.

L9 "is called", I propose the authors rather specify "can be called" or "is called here" (or provide a reference if they use this term from a definition elsewhere).

L13 The authors should provide the reference of where this "has been shown"

l21 What is assumed about the anthropogenic fraction of the Ångström exponent?

L28 "Square" in terms of pixels?

L29 Is the 250x250 pixel square moving with the cloud retrieval? If not, couldn't it easily appear that the nearest aerosol retrieval is in the next, not analysed, square? Maybe the authors can provide a sketch to clarify what exactly they are doing.

L31 Again, it is necessary that the authors define the scale at which they determine a cloud fraction. So far, I understood from the text that they work at the pixel level (1x1

km$^2$). At this scale, cloud fraction is simply zero or one.

P6 l12: ", whereas"

l14: This statement is inconsistent with the "methods" section where it was stated that the aerosol is retrieved at 1 km resolution.

L19: why not describe what actually is found, namely that the Ångström exponent decreases for pixels nearer to the clouds? Of course it is possible to interpret this in terms of particle size, but this cannot be quantified.

L21: this is hard to see from Fig. 2. Could the authors help the reader with a more readable figure, e.g. by horizontal lines?

L23: It should again be made clear what is meant by "contamination". Is the limitation to the inner half of the points in terms of brightness authors the only thing done in the MODIS retrieval to address such issues? Don't they also use different cloud fractions for cloud- and aerosol retrievals?

L25: Of course AOD is also large near clouds due to swelling, but this is not "artificial".

L28: Once more "contamination" - it could be cloud contamination, but could also be 3D effects or swelling, one cannot tell these apart from the analysis.

P7 L28: Obviously the standard error decreases with the sample size as nˆ-1/2. But didn't the authors discuss standard deviation?

P8 L8: To me it seems that the differences are mainly due to the result that for CAPA-L2, the regression coefficients seem to be mostly positive, while for CAPA-2L_15km, there are very large areas where the regression coefficients turn positive.

L12: A better approach would be to show joint histograms. It would be advisable to use the method of Gryspeerdt et al. (2016; doi:10.1002/2015JD023744). It is astonishing that this analysis yields no relationship between cloud albedo and AI, while Fig. 3 and 4 show a substantially positive relationship in the same region. Or do clouds in this

region usually have COD < 5 in the AATSR retrievals?

L23: "Independently derived" seems exaggerated. After all, as I understand, the retrieval algorithm is the same in both cases, as is the way to compute cloud albedo using the radiative transfer model?

P9 l5: Does not Fig. 2 suggest that swelling is negligible at scales > 15 km away from cloud edges?

L18: Which references used in assessment reports do the authors refer to? I'm not aware of many estimates that also include what is called here "extrinsic" forcing.

L26: Is this really the standard error, or not rather the standard deviation of the spatial distribution?

L32: This is only true for some GCMs.

P11, References General comment on references: The authors should consistently show or not show dois and URLs. Journal names should be abbreviated

P14, Caption Fig. 1: 512\times 100 km$^2$

P15, Figure 2: It would be useful to show in addition the product of Ångström exponent and aerosol optical depth. Is AI actually approximately constant with distance from cloud?

P16, Caption Figure 3: It seems what is provided as "mean" is the global mean values? And "Standard deviation" the standard deviation of the spatial variability of the regression coefficients?

P20, Table 1: "CAPA_L2": this seems to correspond to all aerosol retrievals, not just the green ones in Fig.1 "CAPA-L2_15km" I believe these are the green pixels aren't they?

---

## Referee Comment (RC2) · Anonymous Referee #2 · 4 Jul 2017

The paper addresses a very important question, whether using satellite data obtained right near clouds may bias satellite estimates of indirect aerosol radiative forcing. This may occur if complications (such as the presence of cloud drops in supposedly clear areas, aerosol swelling, cloud shadows, or enhanced scattering from clouds into aerosol fields) made aerosol observations near clouds unreliable or unrepresentative. The paper presents a new approach, which avoids these dangers by excluding the potentially compromised aerosol data that was obtained right near clouds. The authors then find that this approach greatly reduces the estimated indirect aerosol radiative forcing values. The overall approach seems reasonable, but I have some significant concerns. The two most important ones are (1) whether random sampling uncertainties have

a large influence on the conclusions, and (2) whether the proposed method yields weaker aerosol-cloud relationships because it uses aerosol data obtained farther away from clouds, where the aerosol population may be less representative of the aerosol population that enters the clouds. Because of these and other concerns, I recommend major revisions to the manuscript.

Most important comments

Page 8, Lines 20-21: It would be important to discuss whether the relationship between cloud and aerosol properties (and so the estimated indirect aerosol radiative forcing values) may be weaker for aerosols farther than 15 km (that is, for CAPA-L2_15km) simply because aerosols farther away are more likely to be in a different air mass (and therefore are not representative of the aerosols that actually interact with the clouds). I am concerned about this, because Line 8 of Page 7 mentions that the median distance between the cloud and aerosol pixels paired up by CAPA-L2_15km is 27 km, which implies even larger distances in some cases.

Figure 7: The large and overlapping error bars raise some questions about the statistical reliability of results. Could it be that the results from CAPA-L2_15km are smaller than the results from the other methods only because of random statistical fluctuations? The similarity of MODIS and AATSR results, and the similar tendencies in Figs. 8 & 9 suggest that the qualitative behavior in Figure 7 is correct despite the large error bars, but it would be important to address the issue of statistical reliability.

Other comments

Page 5, Lines 6-7: Should clarify the definition of c_m_overbar, which is now: "cm is the climatology of low-level clouds having cloud top pressure greater than 500 hPa and composed of liquid phase droplets over ocean regions" to show that the climatology is the climatological mean of \*\*\*cloud cover fraction\*\*\* of liquid water clouds with top pressures exceeding 500 hPa over ocean. Also, it seems best to delete "low-level" from the sentence, as 500 hPa serves as the definition for low levels, and the current wording

could be misunderstood as c_m_overbar telling what fraction of low-level clouds occur below the 500 hPa level (which would imply a remaining fraction of low-level clouds that occurs above the 500 hPa level).

Page 5, Line 10: the first term represents changes not in cloud albedo alone, but in the difference between cloud and clear sky albedos (with clear sky albedo also changing with aerosol loading). So perhaps the word "represents" could be replaced by something like "includes". Or, perhaps even better, the text could specify that the first term represents the fact that aerosol loading (AI) has different impacts on the albedos of cloudy and clear columns. (If it had the same impact on both columns, this term would vanish.)

Page 5, Line 15: At the end of the line, what does F_anth represent and where does it come from? (In other words, how does F_anth relate to tau_anth?)

Page 6, first paragraph: While it is clear why the adopted hybrid approach is faster than the brute force approach (used for high cloud fractions), it would help to also discuss why the hybrid approach is faster than always using the low-cloud-fraction approach.

Page 6: It would help to clarify somewhere, what happens when there are two or more aerosol pixels that are at the same distance from a certain cloud: Does CAPA use only one of these pixels, or does it average the aerosol properties over all of these pixels?

Page 7, line 6: In order to clarify that clouds are paired with aerosols and not with other clouds, I suggest changing "clouds are paired based on the nearest located aerosol (CAPA-L2) and based on the nearest aerosol" to something like "clouds are paired with the nearest located aerosol (CAPA-L2) and with the nearest aerosol…"

Page 7, Line 18: I suggest either clarifying what "1-sigma regression estimate" refers to, or deleting "1-sigma". The same applies to Page 9, Line 11.

Page 8, Line 13: The word "stronger" should be changed to "steeper", which is a more suitable word for describing slopes.

Page 8, Line 24: "MODIS afternoon-train" should be changed to "MODIS Aqua".

Page 18: "CAPA-L2_15km; blue" should be changed to "CAPA-L2_15km; green".

Page 8, Line 31: I suggest replacing "shown in Figure 9" by "shown in Figures 8 and 9"; otherwise the order of the two figures should be reversed (so that Figure 8 is referenced before Figure 9).

Page 9, Lines 8-9: In the sentence "we have reconstructed the pre-averaged aerosol product at first through the removal of near-cloud aerosols in the standard AATSR and MODIS data", it would be important to clarify what is meant by "removal of near-cloud aerosols": Does this mean removing aerosol data for 10X10 km areas that have clouds within 15 km? If so, how was this removal implemented: Was a 10X10 km area removed if any part of it was within 15 km from the nearest cloud? My guess would be that 1 km pixels within 15 km to the nearest cloud were eliminated first, and then the remaining pixels were processed by the 10 km-resolution algorithm.

Page 9, Line 19-20: Does Table 2 show results for all oceans, or does it exclude polar regions or covered by sea ice?

Page 21: Table 2 (along with the lack of CAPA in Figs. 8 & 9) points to an inherent limitation of CAPA: It cannot be used to estimate the extrinsic (or overall) forcing, only the intrinsic forcing. This important limitation of CAPA should be mentioned somewhere prominently, and probably even in the summary or abstract.

Page 10: The text of Section 8 seems to be missing.

Page 10, first line of Section 9: A typo: ATSR should probably be changed to AATSR.

———————————————

---

## Referee Comment (RC3) · Anonymous Referee #3 · 4 Jul 2017

This paper examines how colocation and sampling choices made in aerosol cloud interaction studies impact the strength any aerosol cloud relationships derived in those studies. Using data from MODIS and AATSR, the authors use a nearest neighbour approach to select pairs of aerosol and cloud pixels for analysis. They show that using aerosol retrievals located more than 15km from a cloud reduces the implied aerosol forcing from the AI-cloud albedo relationship as well as the implied extrinsic forcing due to a reduced AI-cloud fraction sensitivity.

The paper is well written and makes an important point about the sampling of aerosol retrievals when used in aerosol-cloud interaction studies. Previous work has shown

that aerosol retrievals are enhanced near clouds, but this work goes further, estimating the impact of this effect on the implied radiative forcing. There are a few points and one algorithmic suggestion that I would make, but other than that it is suitable for publication in Atmospheric Chemistry and Physics.

Main points

I appreciate that the intrinsic forcing concept has been used in the past, but I am not clear that the results from this necessarily carry across to other studies using an aerosol-CDNC relationship to constrain the aerosol indirect forcing. The intrinsic forcing relies on all the properties of the cloud being uncorrelated to CF. However, Feingold et al. (2016) showed that the cloud albedo can be very strongly correlated to CF. Given that cloud properties that are correlated to the CF have the potential to generate spurious aerosol-cloud relationships (Gryspeerdt et al., 2014), this might affect the evaluation of the intrinsic forcing. It is not clear how strongly aerosol-CDNC relationships are affected by covariation with CF, so it would be very interesting to see how the aerosol-CDNC type forcing calculation (e.g. Quaas et al., 2008) is affected by near-cloud aerosol retrievals in addition to the results presented here. I think that this would be necessary if the authors are to apply their conclusions to all satellite-based estimates, rather than just those that are based on the intrinsic forcing concept.

When calculating the radiative forcing, the authors use an anthropogenic aerosol fraction from Bellouin et al., 2013. This fraction is derived from AOD, not AI, and so may be smaller than expected in some regions, especially where dust dominates. There are other possible anthropogenic aerosol produces (such as a fraction from the AeroCom models, Ghan et al., 2016), but it should be noted that this method might produce an underestimate in the forcing through a too small anthropogenic fraction.

While the authors have already produced this dataset, if they wanted to re-run their analysis (or for others who want to reproduce it), it is worth noting that there are much more efficient algorithms for finding nearest neighbours in a large dataset. Binary

search trees, such as a k-d tree or VP tree would work well here. A quick test using the standard python/scipy cKDTree on a close-to-worst-case MODIS granule (MYD06_L2-2013-136-2315, about 40% cloud fraction), found nearest neighbour aerosol pixels for all the cloud pixels (∼1million) in about 1 second with no restrictions on distance. Obviously the algorithm used by the authors can provide extra information, but this might be useful for further work.

Minor points

P2L22: Is it clear that there is a co-location 'problem'? The benefit of CAPA selecting the closest aerosol-cloud pairs is not mentioned beyond increasing the number of samples.

P2L34: How are the aerosols assimilated into models affected by wet deposition (compared to the aerosols retrieved by satellite)? Perhaps this should refer to 'aerosols from reanalysis products'?

P3L6: Does this really provide improved statistics? Many of the retrievals are strongly correlated in space (and time), so it is not clear that more individual datapoints provides more information.

P5L30: Some studies (e.g. Koren et al., 2012) perform interpolation between 1 by 1 degree gridboxes, which is a larger effective scale than the 150km shown here. I don't imagine that limiting the pairs to 150km is much of an issue, but it is not obviously correct to ignore them.

P6L10: Does this interpolation then mean that there are some 10km pixels which are considered both valid and invalid when filtering for pixels 15km from a cloud?

P7L19: As mentioned above, the number of degrees of freedom is smaller than the total number of cloud-aerosol pairs. How is the error estimate then calculated (does it account for the autocorrelation in the cloud and aerosol fields)?

P8L25: 0.1Wm-2 out of 0.4Wm-2 is still quite a large discrepancy

P10L23: The apostrophe in NERCs is not rendering correctly

Fig. 9: These extrinsic forcings (for the corrected L3 data) are quite close to those proposed by Gryspeerdt et al., 2016, which could provide supporting evidence for this proposed extrinsic forcing.

References

Bellouin, N., J. Quaas, J.-J. Morcrette, and O. Boucher (2013), Estimates of aerosol radiative forcing from the MACC re-analysis, Atmospheric Chemistry and Physics, 13(4), 2045–2062, doi:10.5194/acp-13-2045-2013.

Feingold, G., A. McComiskey, T. Yamaguchi, J. S. Johnson, K. S. Carslaw, and K. S. Schmidt (2016), New approaches to quantifying aerosol influence on the cloud radiative effect, Proc Natl Acad Sci U S A, 201514035, doi:10.1073/pnas.1514035112.

Ghan, S. et al. (2016), Challenges in constraining anthropogenic aerosol effects on cloud radiative forcing using present-day spatiotemporal variability, Proc Natl Acad Sci U S A, 201514036, doi:10.1073/pnas.1514036113.

Gryspeerdt, E., P. Stier, and B. S. Grandey (2014), Cloud fraction mediates the aerosol optical depth-cloud top height relationship, Geophys. Res. Lett., 41, 3622–3627, doi:10.1002/2014GL059524.

Koren, I., O. Altaratz, L. A. Remer, G. Feingold, J. V. Martins, and R. H. Heiblum (2012), Aerosol-induced intensification of rain from the tropics to the mid-latitudes, Nature Geoscience, 5, 118, doi:10.1038/ngeo1364.

Quaas, J., O. Boucher, N. Bellouin, and S. Kinne (2008), Satellite-based estimate of the direct and indirect aerosol climate forcing, J. Geophys. Res., 113, 05204, doi:10.1029/2007JD008962.

---

## Author Comment (AC1) · 8 Sep 2017

Responses to reviews of the original submission

Review Comments in black; responses in blue

 **Anonymous Referee #1**

Christensen et al. present a new technique of relating aerosol- and cloud retrievals from satellite data. They created an algorithm that searches for the nearest aerosol retrieval for each cloud retrieval. Different from previous approaches (Bréon et al., 2002), no backtrajectories are computed, but the nearest pixel, independent on whether or

not the aerosol might actually be advected to the cloudy region. Despite this, it is an innovative approach and may indeed help overcome some issues with the aerosol- and cloud retrievals. The authors analyse statistical relationships between the aerosol index and cloud albedo computed on the basis of satellite cloud retrievals using a radiative transfer code, as well as between AI and cloud fraction. They proceed to compute implied radiative forcings.

The manuscript is astonishingly superficial in many of the explanations. Many statements are very difficult to follow, or not at all reproducible from the information provided.

The authors are imprecise in their language. It seems they in general want to assess the effective radiative forcing due to aerosol cloud interactions, i.e. the overall cloud response to the aerosol, including cloud water path and cloud fraction changes.

Nevertheless, it is a useful paper and should eventually be published. However, I have numerous specific points the authors should address.

» Thank you for the excellent and thoughtful feedback. In writing the manuscript, we were aiming for brevity since this is a two-part paper. But you rightly point out that some crucial details are needed in order to reproduce the methodology. Therefore, we have revised the manuscript based on your suggestions and the other reviewers so that this research is understandable and reproducible.

P1 L17 Not so much in satellite estimates

» The relative spread in the effective radiative forcing estimates of aerosol radiation interactions and aerosol cloud interactions (ERFari+aci) based on the "expert judgement" in the IPCC report chapter 7 figure 7.19 has a similar range in both satellite data[-0.93 to -0.45 W/m2 with median value of -0.85 W/m2; (range/median) = 56%] and GCM's [-1.68 to -0.81 with median value of -1.38 W/m2; (range/median) = 63%].

L20 the "buffering" isn't precisely defined. A better more specific explanation on what is missing is necessary

» This sentence has been modified to include "cloud-top entrainment feedbacks in aerosol-cloud interactions" in the description.

P2 L1 given the large range of GCM estimates, it needs to be clarified which publications the authors refer to

» We now reference the appropriate IPCC 5th Assessment Report reference including the specific chapter in which these forcing estimates are published.

L4 remains

» Done

L4-10 the order is awkward. If one had proper CCN retrievals (in the order the authors impose item 5), items 1-3, perhaps even 4, wouldn't matter. Also not all problems are pertinent to all aerosol-cloud interactions. The authors need to be specific about what exactly they want to study and where which of the issues arises.

» Fundamentally, from a satellite perspective we are aiming to obtain a suitable proxy for CCN. It could be argued that points (1-3) hamper attempts to retrieve this accurately from space. I agree that points 4, 5 & 6 are different problems and have therefore deleted them here to make the point clear that we are referring to aerosol retrievals – not aerosol-cloud interaction processes.

L11 the authors need to clarify what they mean by "contamination" (do they mean problem 1, 2, or 3?) 3, to some extent 2, cannot really be called "contamination" since these are plausible physical processes. It is also important that the authors shouldn't forget to mention that clouds are also an actual source of aerosol. Sulfate predominantly nucleates via the aqueous phase.

» We agree that the general word "contamination" is being used too loosely here and throughout the text. Therefore, unless specifically referring to "cloud contamination" in the aerosol product the retrieval is described as being an "artefact" caused by other processes besides humidification which is a real process that enhances the size of the

aerosols near clouds. In this particular sentence we have simply removed the word contamination and replaced it with "retrieval artefacts."

L19 "larger" than what? And do the authors really refer to a forcing here, or rather to an effect?

» Thank you for flagging this the end of the sentence seemed to be missing. The results from the Twohy et al. (2009) refer to the "aerosol direct radiative effect."

L29 that hold of course only if one analyses one grid box over one season. Experience shows that in such attempts, very rarely 90 data points would be available.

» True, when the cloud cover fraction is 100% over a grid-box there is no availale aerosol data and under these conditions this would decrease the number of level-3 pre-averaged samples. Therefore, we have added the words "at most 90-pairs..."

L31 of course also the problem of spurious clouds in pixels labelled cloud-free

» Yes, thank you for pointing this out. This process is now included and discussed.

P3 L1 this statement needs further explanation to be understandable.

» The main point was missed here and restated. Wet deposition processes would be unconstrained in cloudy areas where the satellite cannot retrieve aerosol that is used for assimilation in these reanalysis products.

L3 While the authors call their method "new" they should acknowledge at the presumably first aerosol-cloud interaction study from satellites (Bréon et al., 2002) already applied such a method.

» Thank you for pointing out the method described in Bréon et al. (2002). Their approach is arguably different but indeed another way to pair satellite retrieved aerosols and clouds together. For completeness, we have added a sentence describing the back-trajectory approach and reference their work. Although, it is not clear from Bréon et al. (2002) whether the assumptions are valid, namely, whether errors in the wind field

for which the back-trajectory model relies upon may confound the aerosol-cloud pairing scheme. A deeper examination of this method combined with the screening selection CAPA algorithm, we believe, would be a powerful approach to studying aerosol-cloud interactions. We have therefore included this important step in the manuscript and discuss in the conclusion section how a back trajectory method could used here.

L7 the theoretical maximum for the 1km MODIS retrievals of clouds is about $110 \times 110 \times 90 = 11 \times 10^5$. Is the reduction by a factor of 3-4 an empirical result?

» No, this was a mistake. The calculation was initially based on an estimate using monthly data (30 days) and I forgot to update the value for CAPA based on three months of continuously sampled data. The text has been updated to the correct value of $11 \times 10^5$.

L8 Can the authors clarify what the scale of the MODIS retrievals is? I believe it is 1 km for the cloud product, but is it also 1 km for the aerosol product?

» The MOD04 product is sampled at 10 km spatial resolution but the data has been resampled to 1-km resolution in this study to match the corresponding cloud product which is at 1 km. We have added this pertinent information to the method section of the manuscript.

L16 It would be good to report the overpass time

» This information has now been included for both instruments.

L18 "seconds" should be abbreviated ("s")

» Done

l24 the authors should explain their statement "this consistency is essential". The conclusion is not straightforward, but obviously using the same cloud mask for aerosol and cloud retrievals also introduces issues.

» It could also be argued that by processing visible and IR data simultaneously and

using a single meteorological state, the outputs of ORAC are physically consistent with all radiances observed at TOA. Many other types of algorithms (e.g. MODIS collection 6) process VIS and IR separately and integrating the results implies a physical state that is "inconsistent" with what was observed. ORAC uses the same radiative transfer forward model and particle scattering calculation framework, so systematic errors resulting from those are shared between aerosol and cloud retrievals providing tighter consistency than other retrieval algorithms.

L26 bracket awkward

» This sentence was split into two for increased clarity.

p4 l7 The appropriate reference for MODIS collection 6 cloud products is Platnick et al. (2017, doi:10.1109/TGRS.2016.2610522)

» Thank you for referring us to this recently published paper. It is now cited accordingly to properly reference MODIS C6 products.

l22 "lower" than what?

» This sentence was deleted since active sensing retrievals are not the focus of this work.

L27: $F_{\text{clr}}^{\downarrow}$: why the index "clr", this is just the incident solar radiation, it seems? Why not operate in Eq. 1 simply with reflected fluxes that are actually observed? Also, at a pixel level, CRE is not defined from observations (cloud fraction is either one or zero). At which scale do the authors compute the CRE?

» $F_{\text{clr}}^{\downarrow}$ is the incoming incident solar radiation. This quantity is shown in the equation merely for completeness. We do not operate equation 1 simply with the reflected fluxes because this relationship is strongly influenced by the aerosol effect on cloud fraction. In this sense it is better to decompose it into intrinsic and extrinsic forcings to avoid bias in the strong correlation with cloud fraction Feingold et al. (2016). To clarify, we are not computing CRE at the pixel scale resolution. The derivatives forming equation 4 stem

from two separate populations. 1) the 1-km pixels forming the clear-sky pixels used to compute ($dA_{\text{clr}}/d\ln AI$) and 2) those forming the cloudy-sky pixels ($dA_{\text{cld}}/d\ln AI$). We have added this key piece of information to the text. In a separate paper (Christensen et al. 2016, JGR) we examined these derivatives using pixels containing only cloudy observations but processing the flux algorithm a second time but assuming "clear-sky" conditions and found that the results were equivalent two processing using two separate populations with cloudy and clear-sky pixels. In general, the $dA_{\text{clr}}/d\ln AI$ term is much smaller than $dA_{\text{cld}}/d\ln AI$.

L28: It is a bit misleading to call $F_{\text{obs}}$ "observed". It obviously rather is the flux computed on the basis of the aerosol and cloud retrievals. Why not "all-sky" as usually defined? Are the clear-sky thermodynamic profiles from reanalysis for the appropriate grid cell? Is the humidity for these the all-sky or the clear-sky humidity? In which sense is "clr" less observed than "obs"? Isn't that applying the retrieved aerosols?

» The $F_{obs}$ term is misleading since it isn't actually "observed" like what CERES, for example provides. Therefore, this term has been changed to $F_{allsky}$. The thermodynamic profiles used in the broadband flux calculations are interpolated to each 1-km imager pixel (cloudy or clear) from the N256 spatial resolution of the ECMWF (European Centre for Medium range Weather Forecasting) Interim Reanalysis product. The typical cloud fraction global distribution derived from the allsky composite of low-level warm liquid clouds is provided in Figure 1 of this response, this forms the $c_m$ climatology term for JJA used in Equation 4.

P5 l4 This is a very loose definition of an "indirect effect". The authors can of course define such a quantify. Usually one would call the definition in Eq. 4 something like a "cloud radiative effect sensitivity", and if one multiplies this with the anthropogenic $\Delta AI$, one would obtain a proxy for the effective forcing due to aerosol-cloud interactions (proxy since it only accounts for column physics).

» Good point. The definition now includes the term "cloud radiative effect sensitivity"

L8 it is peculiar that the authors make use here of the annual-mean incoming solar radiation. The factor in the brackets presumably has a strong diurnal and annual cycle. Co-variation of this factor with the incoming solar radiation then leads to possibly substantial differences in the radiative effects compared to the ones proposed by the authors.

» This is a typo/mistake. The incoming solar radiation is not actually based on the annual-mean value. All of the fluxes are instantaneous with the satellite observations but corrected by the daily-mean solar insolation value. To avoid sampling biases we then compute the mean insolation over each season along with the aerosol-cloud sensitivities, first, to limit the co-variation between the annual cloud cycle and the radiative fluxes and then average the seasonal sensitivities together to form the annual mean. We have clarified this point in the text. We follow a similar process of that used by Grandey and Stier (2010)

L9 "is called", I propose the authors rather specify "can be called" or "is called here" (or provide a reference if they use this term from a definition elsewhere).

» The intrinsic/extrinsic forcing concept was first defined in Chen et al. (2014) and now referenced accordingly.

L13 The authors should provide the reference of where this "has been shown"

» Besides the seminal paper by Nakajima et al. (2001) another more recent example describing the use of "Aerosol Index" as a better proxy for CCN than AOD is described in Gryspeerdt et al. (2017).

l21 What is assumed about the anthropogenic fraction of the Ångström exponent?

» In this calculation we have assumed that the fractional change in the anthropogenic AOD is equivalent to the fractional change in anthropogenic AI. However, as pointed out by reviewer 3 this assumption may lead to an underestimation of the aerosol indirect forcing.

L28 "Square" in terms of pixels?  » Yes, the word "pixels" has replaced the word "square".

L29 Is the 250x250 pixel square moving with the cloud retrieval? If not, couldn't it easily appear that the nearest aerosol retrieval is in the next, not analysed, square? Maybe the authors can provide a sketch to clarify what exactly they are doing.  » For clarity we have added the following: "for pixels near the edge of the square region (within 125 km) the search radius is extended into the nearest adjacent square region."

L31 Again, it is necessary that the authors define the scale at which they determine a cloud fraction. So far, I understood from the text that they work at the pixel level (1x1 km$^2$). At this scale, cloud fraction is simply zero or one.

» The cloud fraction for each subsection consisting of 250 x 250 pixels is computed using the 1-km cloud mask data (in MOD06 or ORAC). This clarifying detail has been added.

P6 l12: ", whereas" » Done

l14: This statement is inconsistent with the "methods" section where it was stated that the aerosol is retrieved at 1 km resolution.

» The ORAC aerosol product is retrieved at 1 km resolution. However, the version we use here is averaged over 10 x 10 km$^2$ regions. We have clarified this point in the method section.

L19: why not describe what actually is found, namely that the Ångström exponent decreases for pixels nearer to the clouds? Of course it is possible to interpret this in terms of particle size, but this cannot be quantified.

» ORAC actually retrieves aerosol particle size and we see an increase in particle size in closer proximity to clouds, however, this is not a standard output for the MODIS product so we use Ångström exponent to make a comparison for this plot. To avoid confusion the words "particle size" were removed this statement.

L21: this is hard to see from Fig. 2. Could the authors help the reader with a more readable figure, e.g. by horizontal lines?

» The plot has been modified to plot error bars every other data point to increase visibility.

L23: It should again be made clear what is meant by "contamination". Is the limitation to the inner half of the points in terms of brightness authors the only thing done in the MODIS retrieval to address such issues? Don't they also use different cloud fractions for cloud and aerosol retrievals?

» The MODIS cloud and aerosol products do in fact use different cloud flags, and this in part could be an explanation for why we observe artefacts in collection 6 aerosol near clouds but this would need to be explored further to determine whether it is related to contamination or other processes.

L25: Of course AOD is also large near clouds due to swelling, but this is not "artificial".

» It is "artificial" in the sense that, near clouds, AOD is a less reliable proxy for CCN. Although, this could be caused by many factors so the word "artificial" was removed and we have also included swelling as another mechanism to explain this possible increase.

L28: Once more "contamination" - it could be cloud contamination, but could also be 3D effects or swelling, one cannot tell these apart from the analysis.

» This is a good point which is not being expressed in this paper. The increase in AOD could also be due to 3D effects or swelling near clouds and these processes confound our ability to use the aerosol retrieval as a proxy for CCN.

P7 L28: Obviously the standard error decreases with the sample size as $n^{-1/2}$. But didn't the authors discuss standard deviation?

» This statement is rather obvious from a statistical analysis point of view and does not

actually contribute to the main point in the paragraph so it has been deleted.

P8 L8: To me it seems that the differences are mainly due to the result that for CAPA-L2, the regression coefficients seem to be mostly positive, while for CAPA-2L_15km, there are very large areas where the regression coefficients turn positive.

» Your observation is correct. I have examined this further and found that 11% of the grid-boxes have a negative cloud albedo effect sensitivity in CAPA-L2, whereas 31% are negative in CAPA-L2_15km (see below histogram values for each 1x1 degree region using these composites). This important observation is now stated in the manuscript.

L12: A better approach would be to show joint histograms. It would be advisable to use the method of Gryspeerdt et al. (2016; doi:10.1002/2015JD023744). It is astonishing that this analysis yields no relationship between cloud albedo and AI, while Fig. 3 and 4 show a substantially positive relationship in the same region. Or do clouds in this region usually have COD < 5 in the AATSR retrievals?

» Thank you for recommending the approach used by Gryspeerdt et al. (2016). While we have not tested this approach here we have tested the approach used by Quaas et al. (2008) that is based on changes in cloud droplet number concentration. In general, the relationships yield similar results for the effective aerosol indirect forcing estimate in each of these composites of our dataset (see full response to reviewer #3).

Close inspection of the comparison between Figure 3 and Figure 5 does indeed show that the mean cloud albedo susceptibilities are roughly the same, i.e. providing values of approximately $dA_{\mathrm{cld}}/d\mathrm{ln}AI = 0.1$ for the region off the coast of California. Your point regarding, "it is astonishing that this analysis yields no relationship between cloud albedo and AI" is not accurate. The linear least squares fit regressions for the CAPA-L2 data show a strong relationship and are statistically significant at the 99% confidence interval using a two-tailed t-test. The CAPA-L2_15km shows a very weak sensitivity by comparison, and this is one of the main points of the paper. For reference, the low-level

warm maritime clouds sampled in this region have annual mean COT's roughly around a value of 10 (see plot below).

L23: "Independently derived" seems exaggerated. After all, as I understand, the retrieval algorithm is the same in both cases, as is the way to compute cloud albedo using the radiative transfer model?

» The words "independently derived" were removed. In general, I agree with you that this seems a bit exaggerated. However, the broadband fluxes are computed based on ORAC-AATSR inputs and C6-MODIS inputs so the datasets are independent but the retrievals are not.

P9 l5: Does not Fig. 2 suggest that swelling is negligible at scales > 15 km away from cloud edges?

» I have added to the statement: may be exaggerated due to retrieval artefacts "caused by contamination of the aerosol retrieval, 3D effects and swelling" to this sentence since it is not possible with the analysis tools to attribute a specific mechanism to this response.

L18: Which references used in assessment reports do the authors refer to? I'm not aware of many estimates that also include what is called here "extrinsic" forcing.

» You are correct. The IPCC report does not include the "extrinsic" cloud fraction forcing term. The intrinsic aerosol indirect radiative forcing computed here is consistent (Amiri-Farahani et al., 2016) with the radiative forcing concept used to describe the effective radiative forcing for aerosol-cloud interactions. We therefore specifically point to the "intrinsic" forcing for comparison.

L26: Is this really the standard error, or not rather the standard deviation of the spatial distribution?

» It is the standard deviation of the spatial distribution in Fig 3, this has been clarified in the caption, and the standard one-sigma error propagated from the linear least squares

fit in Figs 7/9.

L32: This is only true for some GCMs.

» In general I would argue this is true for "most" GCMs that were represented in Figure 7.19 in the IPCC report. I appreciate this point, particularly because the GCMs used in the last IPCC report are now more than 5 years old now, so we now reference "most GCMs" instead of the assumed "all GCMs" as previously written.

P11, References General comment on references: The authors should consistently show or not show dois and URLs. Journal names should be abbreviated

» Reference section has been formatted accordingly.

P14, Caption Fig. 1: $512 \times 100 \text{ km}^2$ » Done

P15, Figure 2: It would be useful to show in addition the product of Ångström exponent and aerosol optical depth. Is AI actually approximately constant with distance from cloud?

» Thank you for this suggestion. The relationship between AI and the distance to nearest cloud has been added to this plot which now includes a 3rd row for AI. The relationship is not constant, AI increases nearer to the clouds because the relative changes in AOD are greater than the relative changes in angstrom exponent.

P16, Caption Figure 3: It seems what is provided as "mean" is the global mean values? And "Standard deviation" the standard deviation of the spatial variability of the regression coefficients?

» Yes, this is true. "mean" and "standard deviation of the spatial variability of the regression coefficients" have been added to the caption.

P20, Table 1: "CAPA_L2": this seems to correspond to all aerosol retrievals, not just the green ones in Fig.1 "CAPA-L2_15km" I believe these are the green pixels aren't they?

» Nice catch! This is typo, it should be red and green pixels (not just red).

**References**

Amiri-Farahani, A., Allen, R. J., Neubauer, D., and Lohmann, U.: Impact of Saharan dust on North Atlantic marine stratocumulus clouds: Importance of the semi-direct effect, Atmos. Chem. Phys. Discuss., 2016, 1–25, doi:10.5194/acp-2016-933, 2016.

Bréon, F.-M., Tanré, D., and Generoso, S.: Aerosol Effect on Cloud Droplet Size Monitored from Satellite, Science, 295, 834–838, doi:10.1126/science.1066434, 2002.

Chen, Y.-C., Christensen, M. W., Stephens, G. L., and Seinfeld, J. H.: Satellite-based estimate of global aerosol-cloud radiative forcing by marine warm clouds, Nat. Geosci., 7, 643–646, doi:10.1038/ngeo2214, 2014.

Feingold, G., McComiskey, A., Yamaguchi, T., Johnson, J. S., Carslaw, K. S., and Schmidt, K. S.: New approaches to quantifying aerosol influence on the cloud radiative effect, Proc. Natl. Acad. Sci., 113, 5812–5819, doi:10.1073/pnas.1514035112, 2016.

Grandey, B. S. and Stier, P.: A critical look at spatial scale choices in satellite-based aerosol indirect effect studies, Atmos. Chem. Phys., 10, 11 459–11 470, doi:10.5194/acp-10-11459-2010, 2010.

Gryspeerdt, E., Quaas, J., Ferrachat, S., Gettelman, A., Ghan, S., Lohmann, U., Morrison, H., Neubauer, D., Partridge, D. G., Stier, P., Takemura, T., Wang, H., Wang, M., and Zhang, K.: Constraining the instantaneous aerosol influence on cloud albedo, Proc. Natl. Acad. Sci., doi:10.1073/pnas.1617765114, 2017.

Nakajima, T., Higurashi, A., Kawamoto, K., and Penner, J. E.: A possible correlation between satellite-derived cloud and aerosol microphysical parameters, J. Geophys. Res. Lett., 28, 1171–1174, doi:10.1029/2000GL012186, 2001.

Twohy, C. H., Coakley, J. A., and Tahnk, W. R.: Effect of changes in relative humidity on aerosol scattering near clouds, J. Geophys. Res., 114, doi:10.1029/2008JD010991, D05205, 2009.

[Figure]

**Fig. 1.** Low-level liquid cloud fraction climatology for June, July, August using ORAC AATSR observations over 2008.

[Figure]

Fig. 2. Histogram of the cloud albedo susceptibilities derived over 1x1 degree regions using AATSR ORAC observations annual average over 2002 - 2012 for the aerosol-cloud pairs using the nearest and 15 km pai

**COT (7.40±1.22 )**

| 0.00 | 3.75 | 7.50 | 11.25 | 15.00 |

**Fig. 3.** Mean cloud optical thickness of low-level liquid phase clouds for ORAC AATSR observations annual mean over 2002 - 2012.

---

## Author Comment (AC2) · 8 Sep 2017

Responses to reviews of the original submission

Review Comments in black; responses in blue

**Anonymous Referee #2** The paper addresses a very important question, whether using satellite data obtained right near clouds may bias satellite estimates of indirect aerosol radiative forcing. This may occur if complications (such as the presence of cloud drops in supposedly clear areas, aerosol swelling, cloud shadows, or enhanced scattering from clouds into aerosol fields) made aerosol observations near clouds un-

reliable or unrepresentative. The paper presents a new approach, which avoids these dangers by excluding the potentially compromised aerosol data that was obtained right near clouds. The authors then find that this approach greatly reduces the estimated indirect aerosol radiative forcing values. The overall approach seems reasonable, but I have some significant concerns. The two most important ones are (1) whether random sampling uncertainties have a large influence on the conclusions, and (2) whether the proposed method yields weaker aerosol-cloud relationships because it uses aerosol data obtained farther away from clouds, where the aerosol population may be less representative of the aerosol population that enters the clouds. Because of these and other concerns, I recommend major revisions to the manuscript.

» Thank you for the great feedback, particularly, the main points regarding the statistical sampling. Further analysis of the spatial distribution of the aerosol field was carried out based on your comments. This analysis has increased our confidence that these results are not merely a statistical anomaly.

Most important comments

Page 8, Lines 20-21: It would be important to discuss whether the relationship between cloud and aerosol properties (and so the estimated indirect aerosol radiative forcing values) may be weaker for aerosols farther than 15 km (that is, for CAPA-L2_15km) simply because aerosols farther away are more likely to be in a different air mass (and therefore are not representative of the aerosols that actually interact with the clouds). I am concerned about this, because Line 8 of Page 7 mentions that the median distance between the cloud and aerosol pixels paired up by CAPA-L2_15km is 27 km, which implies even larger distances in some cases.

» This is a valid concern. We have implicitly assumed in this study that the aerosols are fairly homogenous across large spatial scales, up to 150 km according to the results presented in Anderson et al. (2003). However, to corroborate this claim additional tests have been carried out to address the spatial scale dependence of the distance

between the aerosol and cloud data. Using the observations from AATSR we run an additional test in which the aerosol is removed from nearby clouds up to a distance of 30 km and then each cloud is paired to the nearest far-field aerosol pixel at this scale. Overall, the aerosol indirect forcing estimate is somewhat smaller in strength using 30-km scaling ($-0.20 \pm 0.26$ W/m$^2$) compared to the scaling at 15 km ($-0.28 \pm 0.27$ W/m$^2$) but thee differences between the composites are insignificant. This implies that the far-cloud aerosol statistics are representative of the same airmass as those found closer to clouds. This paragraph added to the manuscript.

» Furthermore, we have examined the spatial autocorrelation length scale using a continuous assimilated reanalysis aerosol product, the CAMS model at 0.125 degrees (because it was already available on our system), and find in this dataset the aerosol has a typical spatial autocorrelation length scale of greater than 150 km over 10x10 degree regions in most locations (see Figure 1 in this response below) which is in general agreement with the data presented in Anderson et al. (2003). Therefore, we are confident that in most cases the aerosols located within 15 km of a cloud are highly likely to be in the same airmass as those located farther away up to 150 km in most locations.

Figure 7: The large and overlapping error bars raise some questions about the statistical reliability of results. Could it be that the results from CAPA-L2_15km are smaller than the results from the other methods only because of random statistical fluctuations? The similarity of MODIS and AATSR results, and the similar tendencies in Figs. 8 & 9 suggest that the qualitative behavior in Figure 7 is correct despite the large error bars, but it would be important to address the issue of statistical reliability.

» An argument could be made that the similarity of MODIS and AATSR results suggest that the tendencies are in fact correct despite the relatively large error bars. This is probably likely for two reasons: 1) the standard deviation of the spatial distributions of Figures 4 and Figure 8 would indicate that the spatial maps are quite smooth. It is not the case that in some grid point's there are larger values and in some grid points

smaller values, as one would expect from random fluctuations. In addition, the forcing estimate computed using CAPA-L2_15km (far-field aerosol pairing) decreases for each grid point compared to CAPA-L2 (near-cloud aerosol pairing) and not just in the average. Finally, the decrease for MODIS (with less than 1/4 as many data points) and AATSR is similar which indicate that the results are robust and not random despite the large error bars.

Other comments Page 5, Lines 6-7: Should clarify the definition of c_m_overbar, which is now: "cm is the climatology of low-level clouds having cloud top pressure greater than 500 hPa and composed of liquid phase droplets over ocean regions" to show that the climatology is the climatological mean of ***cloud cover fraction*** of liquid water clouds with top pressures exceeding 500 hPa over ocean. Also, it seems best to delete "low-level" from the sentence, as 500 hPa serves as the definition for low levels, and the current wording could be misunderstood as c_m_overbar telling what fraction of low-level clouds occur below the 500 hPa level (which would imply a remaining fraction of low-level clouds that occurs above the 500 hPa level).

» I agree, the way it is currently written is confusing. "low-level" in this context is redundant anyway so it was removed.

Page 5, Line 10: the first term represents changes not in cloud albedo alone, but in the difference between cloud and clear sky albedos (with clear sky albedo also changing with aerosol loading). So perhaps the word "represents" could be replaced by something like "includes". Or, perhaps even better, the text could specify that the first term represents the fact that aerosol loading (AI) has different impacts on the albedos of cloudy and clear columns. (If it had the same impact on both columns, this term would vanish.)

» The clear-sky albedo change with AI (i.e. $dA_{clr}/d\ln AI$) is small but included in this equation for completeness. It is generally an order of magnitude smaller than the cloudy-sky albedo change with AI so $dA_{cld}/d\ln AI$ essentially represents the strength

of the intrinsic aerosol radiative forcing but this is a good point and have therefore changed this word from "represents" to "includes."

Page 5, Line 15: At the end of the line, what does F_anth represent and where does it come from? (In other words, how does F_anth relate to tau_anth?)

» F_anth is explicitly derived from the MACC-II reanalysis model. This is now stated prominently in the manuscript and included in the syntax for the anthropogenic aerosol fraction equation.

Page 6, first paragraph: While it is clear why the adopted hybrid approach is faster than the brute force approach (used for high cloud fractions), it would help to also discuss why the hybrid approach is faster than always using the low-cloud-fraction approach.

» In general, if the cloud fraction is high but we were to use the low-cloud fraction looping method the algorithm will run slower. This is because there is a higher likelihood of an aerosol pixel located farther away from the cloud (when the cloud fraction is high), and in this case, the algorithm will require more looping around adjacent pixels until a clear-sky pixel is found, therefore the brute force method for this condition. This point has been clarified in the text

Page 6: It would help to clarify somewhere, what happens when there are two or more aerosol pixels that are at the same distance from a certain cloud: Does CAPA use only one of these pixels, or does it average the aerosol properties over all of these pixels?

» Good point to include here! We have added: "if two (or more) aerosol pixels are located at the same distance from the cloud observation then one of them is selected at random."

Page 7, line 6: In order to clarify that clouds are paired with aerosols and not with other clouds, I suggest changing "clouds are paired based on the nearest located aerosol (CAPA-L2) and based on the nearest aerosol? to something like "clouds are paired with the nearest located aerosol (CAPA-L2) and with the nearest aerosol. . ."

» Thanks for noticing this grammatical mistake we have modified the sentence based on your suggestion.

Page 7, Line 18: I suggest either clarifying what "1-sigma regression estimate" refers to, or deleting "1-sigma". The same applies to Page 9, Line 11.

» The "1-sigma regression estimate" refers to the "standard error" of the regression coefficient. This has been added to the text and is now referred to as the "1-sigma standard error regression estimate." This is a standard measurement of the error and describes the accuracy of the linear least squares fit. The uncertainty on the radiative forcing estimate is computed by averaging the 1-sigma standard error regression values over all of the global grid-boxes.

Page 8, Line 13: The word "stronger" should be changed to "steeper", which is a more suitable word for describing slopes.» Done

Page 8, Line 24: "MODIS afternoon-train" should be changed to "MODIS Aqua".

» Correct, the comparison uses MODIS on Aqua and also included the CERES broadband flux data so I have changed this to: "MODIS Aqua and CERES".

Page 18: "CAPA-L2_15km; blue" should be changed to "CAPA-L2_15km; green".

» Nice catch, done.

Page 8, Line 31: I suggest replacing "shown in Figure 9" by "shown in Figures 8 and 9"; otherwise the order of the two figures should be reversed (so that Figure 8 is referenced before Figure 9).

» Done

Page 9, Lines 8-9: In the sentence "we have reconstructed the pre-averaged aerosol product at first through the removal of near-cloud aerosols in the standard AATSR and MODIS data", it would be important to clarify what is meant by "removal of near-cloud aerosols": Does this mean removing aerosol data for 10X10 km areas that have

clouds within 15 km? If so, how was this removal implemented: Was a 10X10 km area removed if any part of it was within 15 km from the nearest cloud? My guess would be that 1 km pixels within 15 km to the nearest cloud were eliminated first, and then the remaining pixels were processed by the 10 km-resolution algorithm.

» The screening approach is very conservative, that is, 10x10 km areas are removed if "any part" of it was within 15 km from the nearest cloud. This important point has been included in the text.

Page 9, Line 19-20: Does Table 2 show results for all oceans, or does it exclude polar regions or covered by sea ice?

» Geographical range was added to the caption $(60°S - 60°N)$. Satellite retrievals are not used if they are over land or sea ice covered regions has also been added to the methodology section for clarity.

Page 21: Table 2 (along with the lack of CAPA in Figs. 8 & 9) points to an inherent limitation of CAPA: It cannot be used to estimate the extrinsic (or overall) forcing, only the intrinsic forcing. This important limitation of CAPA should be mentioned somewhere prominently, and probably even in the summary or abstract.

» Yes, it is true that the output from CAPA itself precludes our ability to compute the extrinsic forcing at the pixel scale. Running CAPA on pixel-scale data is also impractical for most users of this data due to the large data volume required. We have therefore included these points into the summary section but also remind the reader again that CAPA forms an important step in correcting these L3 type products.

Page 10: The text of Section 8 seems to be missing.

» Code to process aerosol, cloud, and broadband fluxes using ORAC can be obtained via https://github.com/ORAC_CC/ORAC.

Page 10, first line of Section 9: A typo: ATSR should probably be changed to AATSR.
» Nice catch. This was changed accordingly.

**References**

Anderson, T. L., Charlson, R. J., Winker, D. M., Ogren, J. A., and Holmen, K.: Mesoscale Variations of Tropospheric Aerosols, J. Atmos. Sci., 60, 119–136, doi:10.1175/1520-0469(2003) 060$<$0119:MVOTA$>$2.0.CO;2, 2003.

a)

e−fold scale aod autocorrelation (214±72.9 km)

b)

e−fold scale = 247 km
lag(1) = 0.98

AOD
autocorrelation

lag/1 (km)

c)

e−fold scale = 303 km
lag(1) = 0.99

AOD
autocorrelation

lag/1 (km)

**Fig. 1.** a) 1/e folding AOD autocorrelation length scale determined using CAMS reanalysis data in 10x10 degree regions. lags plotted b) off the california and c) S. Africa Coasts.

---

## Author Comment (AC3) · 8 Sep 2017

Responses to reviews of the original submission

Review Comments in black; responses in blue

**Anonymous Referee #3**

This paper examines how colocation and sampling choices made in aerosol cloud interaction studies impact the strength any aerosol cloud relationships derived in those studies. Using data from MODIS and AATSR, the authors use a nearest neighbour approach to select pairs of aerosol and cloud pixels for analysis. They show that using

aerosol retrievals located more than 15km from a cloud reduces the implied aerosol forcing from the AI-cloud albedo relationship as well as the implied extrinsic forcing due to a reduced AI-cloud fraction sensitivity.

The paper is well written and makes an important point about the sampling of aerosol retrievals when used in aerosol-cloud interaction studies. Previous work has shown that aerosol retrievals are enhanced near clouds, but this work goes further, estimating the impact of this effect on the implied radiative forcing. There are a few points and one algorithmic suggestion that I would make, but other than that it is suitable for publication in Atmospheric Chemistry and Physics.

» Thank you for the thoughtful and careful review of this manuscript. These are great suggestions and have led to further testing, specifically on relating the intrinsic/extrinsic forcing concepts to other methods which use CDNC relationships.

Main points

I appreciate that the intrinsic forcing concept has been used in the past, but I am not clear that the results from this necessarily carry across to other studies using an aerosol-CDNC relationship to constrain the aerosol indirect forcing. The intrinsic forcing relies on all the properties of the cloud being uncorrelated to CF. However, Feingold et al. (2016) showed that the cloud albedo can be very strongly correlated to CF. Given that cloud properties that are correlated to the CF have the potential to generate spurious aerosol-cloud relationships (Gryspeerdt et al., 2014), this might affect the evaluation of the intrinsic forcing.

» The intrinsic forcing estimate is based on aerosol-cloud susceptibilities where the cloud cover fraction is 100% so there will be no dependence on CF unless the cloud mask is wrong (indicating a cloud where otherwise would cloud-free due to a satellite retrieval failure, but this is unlikely over ocean regions where the surface is "dark"), which for regions over the ocean this is not likely to be a very large concern.

It is not clear how strongly aerosol-CDNC relationships are affected by covariation with CF, so it would be very interesting to see how the aerosol-CDNC type forcing calculation (e.g. Quaas et al., 2008) is affected by near-cloud aerosol retrievals in addition to the results presented here. I think that this would be necessary if the authors are to apply their conclusions to all satellite-based estimates, rather than just those that are based on the intrinsic forcing concept.

» This is an excellent point. And we agree that extending this analysis to more approaches is essential before applying these conclusions to "all" satellite-based estimates. Therefore, we have included additional diagnostics in the output from CAPA and forcings have been computed using the prominent aerosol-CDNC type approach used in Quaas et al. (2008). In this comparison, we focus on the "cloud albedo effect" which is computed using equation (10) of Quaas et al. (2008). The primary difference between this approach and the "intrinsic forcing" equation used here is the product of dlnNd/ dlnAI with the planetary albedo sensitivity term. Using the same initial inputs for liquid cloud fraction, liquid cloud albedo, solar insolation, and anthropogenic aerosol fraction we find the cloud albedo effect from Quaas et al. (2008) provides a somewhat smaller forcing estimate compared to the intrinsic aerosol indirect forcing method the impact of near-cloud aerosols still results in a larger aerosol indirect radiative forcing estimate. For example, the PRE_AVG ($-0.36 \pm 0.32$ W/m$^2$) and CAPA-L2 ($-0.38 \pm 0.22$ W/m$^2$) composites are significantly larger than the PRE_AVG_Corr. ($-0.09 \pm 0.28$ W/m$^2$) and CAPA-L2_15km ($-0.17 \pm 0.19$ W/m$^2$) composites. In conclusion, both methods provide support that the aerosols located near clouds enhance the aerosol-cloud radiative effect compared to the selection of aerosols that are located farther away from clouds. This can be understood from the sensitivity of CDNC to a relative change in AI given by dlnNd/ dlnAI. Figure 1 below shows the global distributions of the CDNC sensitivity for each regime with smaller values associated with the aerosols located farther from clouds.

When calculating the radiative forcing, the authors use an anthropogenic aerosol frac-

tion from Bellouin et al. (2013). This fraction is derived from AOD, not AI, and so may be smaller than expected in some regions, especially where dust dominates. There are other possible anthropogenic aerosol products (such as a fraction from the AeroCom models, Ghan et al. (2016)), but it should be noted that this method might produce an underestimate in the forcing through a too small anthropogenic fraction.

» In this calculation we have assumed that the fractional change in the anthropogenic AOD is equivalent to the fractional change in anthropogenic AI. As you note, this assumption may lead to an underestimation of the aerosol indirect forcing. We have clarified this point in the text and state this assumption prominently. The current MACC product that we use here http://apps.ecmwf.int/datasets/data/cams-climate-forcings/ does not contain an estimate of the anthropogenic aerosol fraction based on the AI calculation. As I understand, later versions of this product will provide the AI-based changes. Our companion paper Neubauer et al (2017), ACP examines this in more detail examining the differences related to the use of changes in AOD VS AI datasets.

While the authors have already produced this dataset, if they wanted to re-run their analysis (or for others who want to reproduce it), it is worth noting that there are much more efficient algorithms for finding nearest neighbours in a large dataset. Binary search trees, such as a k-d tree or VP tree would work well here. A quick test using the standard python/scipy cKDTree on a close-to-worst-case MODIS granule (MYD06_L2-2013-136-2315, about 40% cloud fraction), found nearest neighbour aerosol pixels for all the cloud pixels (about 1million) in about 1 second with no restrictions on distance. Obviously the algorithm used by the authors can provide extra information, but this might be useful for further work.

» We agree there are more advanced and efficient methods to compute nearest neighbour pixels from satellite data using KdTree's, ect. These suggestions are excellent and would highly encourage the use them to others. However, it is beyond the scope of this work to implement these changes now and so have included this information in the summary for others who might like to reproduce the results using these advanced

methods.

Minor points P2L22: Is it clear that there is a co-location 'problem'? The benefit of CAPA selecting the closest aerosol-cloud pairs is not mentioned beyond increasing the number of samples.

» Fundamentally, we want to know about the aerosol within a cloud. We can't obtain this information from these passive sensing instruments, so we have to use measurements somewhere else and give a reasonable argument why they are representative. Our main points are that (a) aerosols near cloud are affected by contamination/humidity/3D-effects and therefore aren't representative, (b) we assume aerosol vary slowly in space, so (c) we think the closest aerosol is a good proxy. The coupling between aerosol and cloud could potentially be improved using a back-trajectory model as was used in Bréon et al. (2002) but this needs to be explored further. We have included this subsequent approach in the paper because we think it could be relevant to future work.

P2L34: How are the aerosols assimilated into models affected by wet deposition (compared to the aerosols retrieved by satellite)? Perhaps this should refer to 'aerosols from reanalysis products'?

» Good point. The impact of wet deposition in the reanalysis model is unconstrained in cloudy areas where the satellite cannot retrieve aerosol. This point has been clarified in the text.

P3L6: Does this really provide improved statistics? Many of the retrievals are strongly correlated in space (and time), so it is not clear that more individual datapoints provides more information.

» The answer depends on the scale of the interaction. At the local-scale (1 km) the AOD retrievals are completely independent from each other but examining the relationship between AOD over some distance we expect spatial autocorrelation. Here, we are essentially measuring AOD several thousand times and so hammering down the

random uncertainty at the local scale. However, as you mention there could be correlation in space (and time). This will depend on the cloud and aerosol type. For some cloud/aerosol types there could be strong correlation but for others maybe not. I would argue that this approach provides potentially more information since it can decrease random uncertainty. However, for (large) correlated cloud/aerosol fields it will probably not provide very much additional information on sampling (see comment P7L19 below). In the response to your next comment (P7L19) we provide an estimate based on a preliminary examination of the autocorrelation spatial scale and determine that the number of samples is not substantially increased over larger spatial scales. As this is an important point we have added in the manuscript that "the number of AOD retrievals are increased but the number of degrees of freedom are likely not significantly increased due to the relatively large spatial autocorrelation length scale of the aerosol optical depth (e.g. Anderson et al. (2003), Schutgens et al. (2016), Kovacs (2006); Santese et al. (2007); Shinozuka and Redemann (2011))."

P5L30: Some studies (e.g. Koren et al. 2012) perform interpolation between 1 by 1 degree gridboxes, which is a larger effective scale than the 150km shown here. I don?t imagine that limiting the pairs to 150km is much of an issue, but it is not obviously correct to ignore them.

» In general, we would not expect the aerosol at lengths greater than 150 km to be very representative of the same airmass as the location of the cloud. As shown in the following papers (e.g. Anderson et al. (2003), Schutgens et al. (2016), Kovacs (2006); Santese et al. (2007); Shinozuka and Redemann (2011)) and our preliminary analysis of CAMS data reported to reviewer #1 comments the typical length-scale of the aerosol field can vary significantly based on location. Note, this length-scale might be longer than that obtained using the raw satellite observations since the analysis is based on the smoothly varying 0.125 degree CAMS reanalysis data. In general, it may be suitable to construct aerosol-cloud pairs that have larger distances than 150 km in some regions (e.g. off the West coast of Africa) but in other regions (Central

Pacific) this would not be a suitable choice as the length scales can be much shorter. A 150 km length-scale seems to be a reasonable threshold based on these spatial autocorrelation studies. Therefore, we would not expect to see a significant benefit in sampling by extending this range to larger than 150 km as most of the aerosol-cloud pairs are established in most regions within the first 50 km.

P6L10: Does this interpolation then mean that there are some 10km pixels which are considered both valid and invalid when filtering for pixels 15km from a cloud?

» The screening approach is very conservative, that is, 10x10 km areas are removed if "any part" of it was within 15 km from the nearest cloud. This important point has been included in the text.

P7L19: As mentioned above, the number of degrees of freedom is smaller than the total number of cloud-aerosol pairs. How is the error estimate then calculated (does it account for the autocorrelation in the cloud and aerosol fields)?

» The uncertainty estimate of the effective radiative forcing is determined by globally averaging all of the 1-sigma error estimates from each region. The 1-sigma standard error estimate in each region is computed from the least squares fit regression coefficient which is affected by the number of degrees of freedom. This can be shown in the following equation

$$\sigma = \sqrt{((SS_y/SS_x) - b^2)/(n^* - 2)} \qquad (1)$$

where, $SS_y$ and $SS_x$ is the ratio of the covariance matrices, and b is the y intercept of the fit-line and $n^*$ is the number of samples measured by the degrees of freedom. As shown by this simple equation n only appears in the denominator of the square root. Therefore, the error estimate becomes larger by $\sqrt{1/n}$ as the number of samples decreases. We have not included the impact of spatial autocorrelation of aerosol and cloud fields in this study but we do expect the strong spatial autocorrelation between aerosol retrievals. This is expected to decrease the number of degrees of freedom.

Although, the ratio of the decrease in DOF would be similar across all of composites based on the formula in Bretherton et al. (1999)

$$n^* = n(1 - r(\Delta t)^2)/(1 + r(\Delta t)^2) \tag{2}$$

where, $r\Delta t$ is the one-lag autocorrelation of the time-series (or spatial direction from origin) with itself, larger values of the autocorrelation decreases $n^*$.

P8L25: 0.1Wm-2 out of 0.4Wm-2 is still quite a large discrepancy

» True, but the 0.1 W/m2 value is about half the size of the 1-sigma standard error on the regression. So we argue that it is within the noise level which is 0.2 W/m2.

P10L23: The apostrophe in NERCs is not rendering correctly » DONE

Fig. 9: These extrinsic forcings (for the corrected L3 data) are quite close to those proposed by Gryspeerdt et al. (2016), which could provide supporting evidence for this proposed extrinsic forcing.

» This key reference has been included in the text.

**References**

Anderson, T. L., Charlson, R. J., Winker, D. M., Ogren, J. A., and Holmen, K.: Mesoscale Variations of Tropospheric Aerosols, J. Atmos. Sci., 60, 119–136, doi:10.1175/1520-0469(2003)060$<$0119:MVOTA$>$2.0.CO;2, 2003.
Bellouin, N., Quaas, J., Morcrette, J.-J., and Boucher, O.: Estimates of aerosol radiative forcing from the MACC re-analysis, Atmos. Chem. Phys., 13, 2045–2062, doi:10.5194/acp-13-2045-2013, 2013.
Bréon, F.-M., Tanré, D., and Generoso, S.: Aerosol Effect on Cloud Droplet Size Monitored from Satellite, Science, 295, 834–838, doi:10.1126/science.1066434, 2002.
Bretherton, C. S., Widmann, M., Dymnikov, V. P., Wallace, J. M., and Blade, I.: The Effective

Number of Spatial Degrees of Freedom of a Time-Varying Field, Journal of Climate, 12, 1990–2009, doi:10.1175/1520-0442(1999)012<1990:TENOSD>2.0.CO;2, 1999.

Feingold, G., McComiskey, A., Yamaguchi, T., Johnson, J. S., Carslaw, K. S., and Schmidt, K. S.: New approaches to quantifying aerosol influence on the cloud radiative effect, Proc. Natl. Acad. Sci., 113, 5812–5819, doi:10.1073/pnas.1514035112, 2016.

Ghan, S., Wang, M., Zhang, S., Ferrachat, S., Gettelman, A., Griesfeller, J., Kipling, Z., Lohmann, U., Morrison, H., Neubauer, D., Partridge, D. G., Stier, P., Takemura, T., Wang, H., and Zhang, K.: Challenges in constraining anthropogenic aerosol effects on cloud radiative forcing using present-day spatiotemporal variability, PNAS, 113, 5804–5811, doi:10.1073/pnas.1514036113, 2016.

Gryspeerdt, E., Stier, P., and Grandey, B. S.: Cloud fraction mediates the aerosol optical depth-cloud top height relationship, J. Geophys. Res. Lett., 41, 3622–3627, doi:10.1002/2014GL059524, 2014.

Gryspeerdt, E., Quaas, J., and Bellouin, N.: Constraining the aerosol influence on cloud fraction, J. Geophys. Res., 121, 3566–3583, doi:10.1002/2015JD023744, 2015JD023744, 2016.

Kovacs, T.: Comparing MODIS and AERONET aerosol optical depth at varying separation distances to assess ground-based validation strategies for spaceborne lidar, Journal of Geophysical Research: Atmospheres, 111, n/a–n/a, doi:10.1029/2006JD007349, http://dx.doi.org/10.1029/2006JD007349, d24203, 2006.

Quaas, J., Boucher, O., Bellouin, N., and Kinne, S.: Satellite-based estimate of the direct and indirect aerosol climate forcing, J. Geophys. Res., 113, D05 204, doi:10.1029/2007JD008962, 2008.

Santese, M., De Tomasi, F., and Perrone, M. R.: Moderate Resolution Imaging Spectroradiometer (MODIS) and Aerosol Robotic Network (AERONET) retrievals during dust outbreaks over the Mediterranean, Journal of Geophysical Research: Atmospheres, 112, doi:10.1029/2007JD008482, d18201, 2007.

Schutgens, N. A. J., Gryspeerdt, E., Weigum, N., Tsyro, S., Goto, D., Schulz, M., and Stier, P.: Will a perfect model agree with perfect observations? The impact of spatial sampling, Atmospheric Chemistry and Physics, 16, 6335–6353, doi:10.5194/acp-16-6335-2016, https://www.atmos-chem-phys.net/16/6335/2016/, 2016.

Shinozuka, Y. and Redemann, J.: Horizontal variability of aerosol optical depth observed during the ARCTAS airborne experiment, Atmospheric Chemistry and Physics, 11, 8489–8495, doi:10.5194/acp-11-8489-2011, 2011.

[Figure]

[Figure]

**Fig. 1.** Log change in droplet number concentration to a relative change in aerosol index for a) pre-averaged, b) CAPA near-cloud aerosol, and c) CAPA_15 km far-field aerosol using 2002-2012 AATSR ORAC.